

# Comparison of three aerosol chemical characterization techniques utilizing PTR-ToF-MS: A study on freshly formed and aged biogenic SOA

Giorgos I. Gkatzelis [1], Ralf Tillmann[1], Thorsten Hohaus [1], Markus Müller [2,4], Philipp Eichler [2], Kang-Ming Xu [3], Patrick Schlag [1], Sebastian H. Schmitt [1], Robert Wegener[1], Martin Kaminski[1], Rupert Holzinger [3], Armin Wisthaler [2,5], Astrid Kiendler-Scharr [1]

[1] Institute of Energy and Climate Research, IEK-8: Troposphere, Forschungszentrum Jülich GmbH, Jülich, Germany
[2] Institut für Ionenphysik und Angewandte Physik, Universität Innsbruck, Innsbruck, Austria
[3] Institute for Marine and Atmospheric research Utrecht, Princetonplein 5, 3584 CC, Utrecht, The Netherlands
[4] Ionicon Analytik GmbH, Innsbruck, Austria
5 Department of Chemistry, University of Oslo, Norway

*Correspondence to*: R. Tillmann (r.tillmann@fz-juelich.de)

## Abstract

An inter-comparison of different aerosol chemical characterization techniques has been performed as part of a chamber study of biogenic SOA formation and aging at the atmosphere simulation chamber SAPHIR. Three different aerosol sampling techniques, the aerosol collection module (ACM), the chemical analysis of aerosol on-line (CHARON) and the collection thermal desorption unit (TD) were connected to Proton Transfer Reaction Time of Flight Mass Spectrometers (PTR-ToF-MS) to provide chemical characterization of the SOA. The techniques were compared among each other and to results from an Aerosol Mass Spectrometer (AMS) and a Scanning Mobility Particle Sizer (SMPS). The experiments investigated SOA formation from the ozonolysis of $\beta$-pinene, limonene, a $\beta$-pinene/limonene mix and real plant emissions from *Pinus sylvestris L.* (Scots pine). The SOA was subsequently aged by photooxidation except for limonene SOA which was aged by $NO_3$ oxidation.

Despite significant differences in the aerosol collection and desorption methods of the PTR based techniques, the determined chemical composition, i.e. the same major contributing signals were found by all instruments for the different chemical systems studied. These signals could be attributed to known products expected from the oxidation of the examined monoterpenes. The sampling and desorption method of ACM and TD, provided additional information on the volatility of individual compounds and showed relatively good agreement.

Averaged over all experiments, the total aerosol mass recovery compared to an SMPS varied from 80 ± 10%, 51 ± 5% and 27 ± 3% for CHARON, ACM and TD, respectively. Comparison to the oxygen to carbon ratios (O:C) obtained by AMS showed that all PTR based techniques observed lower O:C ratios indicating a loss of molecular oxygen either during aerosol sampling or detection. The differences in total mass recovery and O:C between the three instruments resulted predominately from differences in the field strength (E/N) in the drift-tube reaction ionization chambers of the PTR-ToF-MS instruments and from dissimilarities in the collection/desorption of aerosols. Laboratory case studies showed that PTR-ToF-MS E/N conditions influenced fragmentation which resulted in water loss and carbon-oxygen bond breakage of the detected molecules. Since ACM and TD were





operated in higher E/N compared to CHARON this resulted to higher fragmentation, thus affecting primarily the

detected oxygen and carbon content and therefore also the mass recovery. Overall, these techniques have been

shown to provide valuable insight on the chemical characteristics of BSOA, and can address unknown

thermodynamic properties such as partitioning coefficient values and volatility patterns down to a compound

specific level.

## 1  Introduction

Atmospheric organic aerosols (OA) represent a major contribution to the submicrometer particulate matter ($PM_1$)

thus playing a key role in climate change and air quality (Kanakidou et al., 2005). OA are either directly emitted

through e.g. combustion processes (primary OA, POA) or formed through the oxidation of volatile organic

compounds (VOCs), called secondary OA (SOA) (Seinfeld and Pandis, 2006). SOA constitute a major fraction of

OA (Jimenez et al., 2009) with biogenic VOC oxidation products affecting their global contribution (Guenther et al.,

2012). Due to thousands of individual compounds involved in SOA, the chemical characterization of OA still

presents a huge analytical challenge (Goldstein and Galbally, 2007). The ability of these compounds to condense to

the particulate-phase or partition between the gas and particle phase as well as their volatility are thermodynamic

parameters of interest that determine their atmospheric fate.

Various techniques have been established in order to better quantify and chemically characterize SOA (Hallquist et

al., 2009). These techniques optimize and compromise for time, size or chemical resolution combined with the

percentage of OA mass they can detect. Off-line techniques, based on filter measurement, provide detailed

information on functional groups or individual chemical species while having low time resolution (hours to days)

and size information. On-line techniques, like e.g. the Aerodyne aerosol mass spectrometer (AMS) (Canagaratna et

al., 2007), provide high time resolution and size resolved data while less specific chemical composition information

or molecular identification of the OA compounds is acquired.

In recent years attempts to develop new techniques that combine both chemical identification but also improved

time resolution have been established. These techniques use different pre-concentration methods in order to detect

the particulate-phase compounds. Filter based techniques like the Filter Inlet for Gases and AEOROsols

(FIGAERO) (Lopez-Hilfiker et al., 2014) provide highly effective collection of particles on filters, under high flow

rates (30 standard Liters per minute, sLpm), thus low collection times. Thermal desorption of the sampled particles

on the filter is performed with the disadvantage of sampling artefacts from gas-phase compounds that may condense

on the large surface area of the filter and contribute to the overall signal. Other techniques, like the thermal

desorption aerosol gas chromatograph (TAG) (Kreisberg et al., 2009; Williams et al., 2006) or the collection thermal

desorption unit (TD) (Holzinger et al., 2010b), utilize the concept of particle collection on an impaction surface by

means of humidification and inertial impaction, followed by desorption. TAG and TD provide hourly time

resolution measurements, and when combined with a gas-phase denuder avoid sampling of additional gas-phase

constituents on their collection thermal desorption (CTD) cell. Due to the particle humidification step these

techniques may bias collection efficiency towards water soluble compounds. The aerosol collection module (ACM)

(Hohaus et al., 2010) collects aerosols by passing them through an aerodynamic lens for particle collimation (Liu et





al., 1995a; b), further through a vacuum system (comparable in design to the AMS), and finally impacting the particle phase on a cooled sampling surface. Although the ACM has a low time resolution (3-4 h), it's design makes it applicable for the investigation of compound specific thermodynamic properties e.g. partitioning coefficient and volatility (Hohaus et al., 2015). The chemical analysis of aerosol online (CHARON) (Eichler et al., 2015) is a technique that provides on-line real time measurements by passing the particles through a denuder to strip off the gas-phase. Particles are sampled through an aerodynamic lens combined with an inertial sampler for the particle-enriched flow, and a thermodesorption unit for particle volatilization prior to chemical analysis. The enrichment factor of this system is known by performing calibrations, thus reducing the quantification uncertainty. All the above pre-concentration systems detect the compounds originating from the particulate-phase that underwent evaporation to the gas-phase by desorption, thus introducing possible thermal break down of analytes.

A variety of detection instruments have been coupled to these inlet techniques, providing different functionality and chemical composition information. The proton transfer reaction time of flight mass spectrometer (PTR-ToF-MS) (Jordan et al., 2009) is a soft ionization technique with low detection limits and high time resolution (ms), that can cover a wide volatility range, from VOCs to low VOCs (LVOCs), depending on the inlet used (Eichler et al., 2017). Techniques utilizing a PTR-ToF-MS are capable of measuring a large fraction of the OA mass, ranging from 20 to 100% (Eichler et al., 2015; Mensah et al., 2012), and provide additional information on the elemental composition of the organic compounds; however, the compound's molecular identity attribution is challenging. On the contrary, gas chromatography mass spectrometry is considered ideal for detailed compound specific structural analysis. Techniques like the TAG have been applied utilizing a gas chromatograph, to provide non-polar and low-polarity tracers identification while the modified semi-volatile TAG (SV-TAG) has broadened this range to highly polar oxygenates, mostly seen in the atmosphere, by using online derivatization (Isaacman et al., 2014; Zhao et al., 2013). The volatility and polarity separator (VAPS) is a similar technique that provides volatility- and polarity-resolved OA information by using a modified 2-dimensional gas chromatography (2D-GC) approach combined with high resolution time -of -flight mass spectrometry (Martinez et al., 2016). Although these techniques provide chemical speciation and lower time resolution, they can only do so for a small fraction of the OA mass (10 - 40%).

The specificity of the above newly developed techniques is still to be explored in detail. In this work, an inter-comparison campaign was performed in the atmosphere simulation chamber SAPHIR (Rohrer et al., 2005) to investigate biogenic SOA (BSOA) formation and aging. The focus of this work is on the comparison of three different aerosol characterization techniques, the ACM - PTR-ToF-MS, the TD - PTR-ToF-MS and the CHARON - PTR-ToF-MS. The OA mass fraction these techniques were able to detect combined with the OA chemical characteristics and volatility trends were investigated and compared.

## 2   Methods and instrumentation

### 2.1   Facilities

Experiments were conducted in the atmospheric simulation chamber SAPHIR (Simulation of Atmospheric PHotochemistry In a large Reaction chamber) located in Jülich, Germany. The chamber consisted of twin FEP Teflon foils with a volume of 270 m$^3$, resulting in a surface to volume ratio of approximately 1 m$^{-1}$. High purity



nitrogen (99.9999% purity) was flushed at all times to the space between the twin walls and a pressure gradient was

maintained in order to prevent contamination from outside. A high flow (150 to 200 m$^3$ h$^{-1}$) of air was introduced in order to clean the chamber and reach aerosol and trace gases concentrations below detection limits before each
experiment was initiated. A low flow (8 m$^3$ h$^{-1}$) was used to replenish SAPHIR during experiments from losses due to leaks and sampling of the instruments. The chamber is equipped with a louvre system thus experiments could be
performed under dark conditions focusing on O$_3$ and NO$_3$ oxidation (roof closed) or as photooxidation experiments utilizing sun light (roof open). More details on SAPHIR can be found in Rohrer et al. (2005).
A PLant chamber Unit for Simulation (PLUS) was recently coupled to SAPHIR to investigate the impact of real plant emissions on atmospheric chemistry (Hohaus et al., 2016). PLUS is an environmentally controlled, flow
through plant chamber where continuous measurements and adjustments of important environmental parameters (e.g., soil relative humidity, temperature, photosynthetical active radiation) are performed. To simulate solar
radiation and control the tree emissions in PLUS, 15 light-emitting diode (LED) panels were used with an average photosynthetically active radiation value (PAR) of 750 nm and an average temperature of 25 °C . In this study,
BVOC emissions were generated from 6 *Pinus sylvestris L.* (Scots pine) trees.

A set of standard instrumentation was coupled to the simulation chamber SAPHIR. Air temperature was measured

by an ultrasonic anemometer (Metek USA-1, accuracy 0.3 K) and humidity was determined with a frost point hygrometer (General Eastern model Hygro M4). NO and NO$_2$ measurements were performed with a
chemiluminescence analyser (ECO PHYSICS TR480) equipped with a photolytic converter (ECO PHYSICS PLC760). Ozone was measured by an UV absorption spectrometer (ANSYCO model O341M). Particle size
distribution was measured using a Scanning Mobility Particle Analyser (SMPS TSI, TSI Classifier model 3080, TSI DMA 3081, TSI Water CPC 3786), measuring in the 40 - 600 nm range with a time resolution of 8.5 min and an
accuracy of 12% (Wiedensohler et al., 2012). A High-Resolution Time-of-Flight Aerosol Mass Spectrometer (HR-ToF-AMS) (Canagaratna et al., 2007; DeCarlo et al., 2006) was used to determine the total organic mass and
composition of the SOA formed with an accuracy of 31% (Aiken et al., 2008). High resolution mass spectra were analyzed using the software packages SQUIRREL (v1.57) and PIKA (v1.15Z). Oxygen to carbon ratios were
calculated based on the newly developed "Improved-Ambient" method by Canagaratna et al. (2015).

## 2.2 Experimental procedure

SOA was formed through the ozonolysis of different monoterpenes using the simulation chamber SAPHIR. Experimental starting conditions varied from the injection of β-pinene and limonene, as single compounds or as a
mixture, to the injection of real plant emissions from 6 Pinus sylvestris *L.* (Scots pine), provided from SAPHIR-PLUS (Section 2.1). For the tree emissions experiment the BVOCs consisted of 42% δ$^3$-carene, 38% α-pinene, 5%
β-pinene, 4% myrcene, 3% terpinolene and 8% other monoterpenes, as determined by GC-MS measurements. The details of the experiments are given in Table 1. The chamber was initially humidified (55% RH, 295 – 310 K) and
background measurements for all instruments were performed. Monoterpenes were injected either with a Hamilton syringe injection and subsequent evaporation into the replenishment flow of SAPHIR, or by SAPHIR-PLUS (real
tree emissions). After one hour, ozone was introduced in the system to initiate chemistry. The ozonolysis of





monoterpenes and the tree emissions where performed under low $NO_x$ conditions (10 – 60 pptV) in the absence of

an OH scavenger. For the limonene experiment, 8 hours after the ozone injection, an addition of 30 ppbV of NO was

introduced into the dark chamber. The reaction of NO with remaining ozone in the chamber resulted in the

generation of $NO_3$, thus initiating the $NO_3$ oxidation chemistry. In all other experiments the chamber was

illuminated 20 hours after the ozone injection, exposing the SOA to real sunlight, thus initiating photo-oxidation by

OH radicals. Finally, for the real tree emissions, after 11 hours of ozone exposure, additional biogenic VOCs

(BVOCs) were re-introduced into the SAPHIR chamber to generate fresh SOA which was subsequently aged by

photooxidation for additional 6 hours. The duration of the experiments varied from 17 to 36 hours, providing ample

time to experimentally investigate the aging of the biogenic SOA.


### 2.3    PTR-ToF-MS aerosol chemical characterization techniques

Three independent aerosol chemical characterization techniques utilizing PTR-ToF-MS were compared, the aerosol

collection module (ACM – PTR-ToF-MS, referred to as "ACM" hereafter), the chemical analysis of aerosol online

(CHARON – PTR-ToF-MS, referred to as "CHARON" hereafter) and the collection thermal desorption unit (TD –

PTR-ToF-MS, referred to as "TD" hereafter). Their characteristics and differences are provided in Table 2 and

discussed in detail in this section. The time resolution of the techniques varied from CHARON providing online

measurements to the TD and ACM having increased collection times of 120 and 240 min, respectively. CHARON

was operated at a constant temperature and lower pressure (< 1 atm) while ACM and TD, operated at 1 atm,

introduced temperature ramps during desorption thus providing more detailed volatility information. The limit of

detection (LOD), dependent on the different pre-concentration factors for each technique, resulted in TD having the

lowest LOD (0.001 ng m$^{-3}$), followed by the CHARON (1.4 ng m$^{-3}$), while ACM showed the highest values

(250 ng m$^{-3}$). Different electric field strength (V cm) to buffer gas density (molecules cm$^{-3}$) ratio (E/N) conditions

were applied to the PTR-ToF-MS of each aerosol chemical characterization technique. Lower E/N set values

resulted in longer ion residence times in the drift tube of the PTR-ToF-MS thus higher sensitivity due to enhanced

proton transfer reaction times. Ions were introduced to a lower kinetic energy system, thus resulting in reduced

fragmentation during ionization while the cluster ion distribution was changed when lowering the E/N, supporting

more $H_3O^+(H_2O)_n$ (n=1,2,3..) cluster ion generation (de Gouw and Warneke, 2007). Since the proton affinity of

$H_3O^+(H_2O)_n$ is higher than that of $H_3O^+$, a certain range of organic compounds could not be ionized in such

operating conditions. Based on the uncertainty in the reaction rate coefficient of the organic compounds with $H_3O^+$

the PTR-ToF-MS was assumed to introduce a ± 40% uncertainty on the volume mixing ratios of uncalibrated

compounds for CHARON and TD. The ACM used an average sensitivity of 15 ncps/ppbV with an uncertainty of

± 50% (± 1σ).

#### 2.3.1    ACM – PTR-ToF-MS

The ACM is an aerosol collection inlet with subsequent sample evaporation coupled to a gas-phase detector

designed for in situ, compound specific chemical analysis. The ACM can be adapted to work with different gas-



phase analysers and has previously been used coupled to a GC-MS (Hohaus et al., 2010). In this work, the ACM
   was coupled to a PTR-ToF-MS (model PTR-TOF 8000; Ionicon Analytik GmbH, Innsbruck, Austria).

In brief, ambient air was sampled through an aerodynamic lens (Liu et al., 1995a; b) with a flow rate of 80 ml min$^{-1}$.
   Within the aerodynamic lens the gas and particle phase of an aerosol were separated and the particles were

collimated into a narrow beam. The particle beam was directed through a high vacuum environment ($10^{-5}$ torr) to a
   cooled (-5 °C) sampling surface made of Siltek®/Sulfinert®-treated stainless steel. After collection was completed

(a collection time of 4 h was used in this study) the particles were thermally desorbed by heating up the collector.
   The evaporated compounds were transferred to the PTR-ToF-MS through a coated stainless steel line of 0.8 mm

inner diameter and 30 cm length, constantly kept at 300 °C. Nitrogen was used as carrier gas with a flow of
   300 ml min$^{-1}$, resulting in a residence time of 60 ms. For this study, the collector temperature was ramped by

100 °C min$^{-1}$ to a maximum of 250 °C, with 3-minute isothermal sections at 100 °C, 150 °C and 250 °C,
   respectively. During the final temperature step of 250 °C, desorption time was extended for additional 7 minutes to

ensure complete evaporation of the sample. These temperature steps provided enough time for compounds to
   undergo evaporation within a defined volatility range. The signal dropped close to zero before each temperature step

was completed, making the ACM-PTR-ToF-MS ideal for compound specific volatility trend analysis. Parallel to the
   ACM particulate-phase collection, a bypass line was used, coupled to the same PTR-ToF-MS, measuring the gas-

phase during particle phase sampling time. An example of the gas and particulate-phase measurements is given in
   Figure S1. During the campaign, the aerosol-phase sampling line was a stainless steel line

(total length: 4 m, OD: 1/4') with a flow of 0.7 L min$^{-1}$.

   Assuming a collection efficiency of 100% (Hohaus et al., 2010) for all particles in the aerosol sample, measured

PTR-ToF-MS signals could be converted to particulate mass concentrations by applying PTR calibrations as
   described in the following. Normalization of the PTR-ToF-MS counts per second was performed based on the $H_3O^+$

signal, resulting in ncps. The ACM was corrected for mass discrimination. The mass discrimination function was
   determined based on the ratio of the measured over the theoretical sensitivity of acetaldehyde, acetone, butanone,

benzene, toluene, xylene and mesitelyne. The instrument was calibrated for a total of 15 compounds including
   aromatics (benzene, toluene, xylene, chlorobenzene), oxygenates (acetaldehyde, acetone, 2-butanone, 3-pentanone,

MVK, nopinone, methanol, 1-butanol), pure hydrocarbons (isoprene, α-pinene) and acetonitrile. Calibration was
   performed by coupling the PTR-ToF-MS to a calibration unit (LCU, Ionicon Analytik GmbH, Innsbruck, Austria)

and measuring known concentration of the compounds in the gas-phase. For signals observed at uncalibrated masses
   the average sensitivity of acetaldehyde, acetone, MVK, Butanon, pentanone and nopinone was applied resulting in

15 ncps/ppb. The mass concentration of an aerosol compound $z_i$ in the air sample was calculated based on the
   mixing ratios the PTR-MS measures:

$$\mathbf{mz}_{i,(\mu g/m^3)} = \frac{mz_{i,(ppb)} \times MW_i}{T \times R} \times \frac{F_{N_2} \times t_{meas}}{F_{col} \times t_{col}}, \qquad\qquad (1)$$

   where $mz_{i,(\mu g/m^3)}$ is the aerosol concentration of compound i in µg m$^{-3}$, $mz_{i,(ppb)}$ the background corrected

arithmetic mean of the mixing ratio during the aerosol analysis in the nitrogen flow in ppb, $MW_i$ is the molecular
   weight of compound i in gmol$^{-1}$ −, R is the universal gas law constant, T the ambient temperature of the SAPHIR




chamber in Kelvin, $F_{N_2}$ the flow of the carrier gas in standard liter per minute, $t_{meas}$ the aerosol desorption duration, $F_{col}$ the collection flow rate of the aerosol to the ACM in standard liter per minute and $t_{col}$ the aerosol collection
duration. The volume ratio correction $\left(\frac{F_{N_2} \times t_{meas}}{F_{col} \times t_{col}}\right)$ was applied in order to account for the ACM collection preconcentration step. The mass concentration was calculated by taking into account only the signal above the
instrument noise (> 2σ) for each compound at each desorption.

Background measurements were performed before and after every experiment (~ 2 times per day) by heating up the

collector, without depositing particles on the surface beforehand. The signal derived from the background measurements at each temperature step was then interpolated and subtracted from all desorptions for all compounds.
Two major factors could affect the background signal, gas-phase interference and aerosol residual remaining at the collector after each desorption cycle. Due to the aerodynamic lens set-up the ACM design prevents gas-phase
contamination (removal > 99.9999%). Background measurements throughout this study show no residual compounds on the collector in the desorption temperature range studied.
PTR-ToF-MS operation conditions were kept constant throughout the campaign. It was operated at E/N = 120 Td. The drift tube was kept at a temperature of 100 °C and a pressure of 2.30 mbar. The mass resolving power of this
PTR-ToF-MS was m/Δm ~ 2500 (Δm is full width at half maximum). Mass spectra were collected up to *m/z* 400 at 10 s signal integration time. Analysis of the raw data was performed using the PTR-TOF Data Analyzer (version
4.40) software (Müller et al., 2013). In brief, an integration time of 90 s was chosen for the software and *m/z* calibration peaks were assigned based on the peaks of 21.02, 59.05 and 180.94 accounting for $H_3[^{18}O]^+$, protonated
acetone and trichlorobenzene respectively. Trichlorobenzene was used as an internal standard throughout the campaign. The chemical composition assignment was derived from the measured exact mass assuming a molecular
formula of $C_xH_yO_zN_a$ and attributing the isotopic pattern when possible.

### 2.3.2    CHARON – PTR-ToF-MS

The analyzer deployed by the University of Innsbruck consisted of a Chemical Analysis of Aerosol Online (CHARON) inlet interfaced to a PTR-ToF-MS.
The CHARON inlet (Eichler et al., 2015) consists of a gas-phase denuder for stripping off gas-phase analytes, an aerodynamic lens for particle collimation combined with an inertial sampler for the particle-enriched flow, and a
thermodesorption unit for particle volatilization prior to chemical analysis. The monolithic charcoal denuder (Mast Carbon International Ltd., Guilford, UK) used in this study was 25 cm long, had an outer diameter of 3 cm and a
channel density of 585 channels per inch (cpi). The thermodesorption unit consisted of a heated Siltek®/Sulfinert®-treated stainless steel tube kept at a temperature of 140 °C and a pressure on the order of a few mbar. A HEPA filter
(ETA filter model HC01-5N-B, Aerocolloid LLC, Minneapolis, MN, USA) was periodically placed upstream of the gas-phase denuder for determining the instrumental background. More details on the performance of the CHARON
inlet are given in Eichler et al. (2015).

The CHARON inlet was interfaced to a commercial PTR-ToF-MS instrument (model PTR-TOF 8000; Ionicon

Analytik GmbH, Innsbruck, Austria). PTR-ToF-MS mass spectra were collected up to *m/z* 500 at 10 s signal

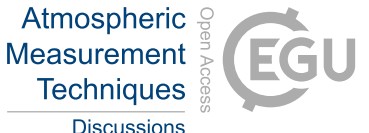

integration time. The PTR-TOF Data Analyzer (version 4.40) software was used for data analysis (Müller et al.,

2013). During the tree emissions experiment the electric field applied to the drift tube was periodically switched in

300 s intervals, i.e. measurements were performed at alternating E/N-values of 65 Td (referred to as "CHARON$_{65}$"

hereafter) and 100 Td (referred to as "CHARON$_{100}$" hereafter), respectively (1 Td = $10^{-17}$ V cm$^{-2}$ molecule$^{-1}$). For all

other experiments the E/N-value analysed was at 100 Td. The drift tube was kept at a temperature of 120 °C and a

pressure of 2.40 mbar. Continuous permeation of 1,2-diiodobenzene was performed into the drift tube for generating

mass axis calibration signals at *m/z* 203.943 and *m/z* 330.847. The PTR-ToF-MS was characterized using a 16-

compound gas mixture that included aromatics (benzene, toluene, o-xylene, mesitylene, chlorobenzene), oxygenate

compounds (acetaldehyde, acetone, 2-butanone, 3-pentanone, MVK, nopinone, methanol, 1-butanol), pure

hydrocarbons (isoprene, α-pinene) and acetonitrile. The mass resolving power of this PTR-ToF-MS was m/Δm

4500-5000.

The entire CHARON setup was calibrated using size-selected ammonium nitrate particles as described in Eichler et

al. (2015). A sensitivity model based on Su and Chesnavich's parameterized reaction rate theory and a chemical

composition based parameterization of polarizabilities at a constant dipole moment of $\mu_D$ = 2.75 D (between 1 – 4.5

D for most oxygenated organic compounds) was applied to calculate sensitivities of unknown compounds. This

resulted in an *m/z* independent sensitivity accuracy of about ± 25%. For compounds without assigned elemental

composition the polarizability of acetone was applied with an accuracy of ± 40%. Derived volume mixing ratios

were transformed to mass concentrations using the molecular *m/z* information at Normal Temperature and Pressure

(NTP) conditions (293.15 K, 101.325 kPa). Quantification was hampered by two events (power failure, partial

obstruction of the aerodynamic lens) which resulted in a higher than usual variability of the particle enrichment in

the aerodynamic lens. Results from two experiments (limonene ozonolysis/NO$_3$ oxidation and limonene/β-pinene

mixture ozonolysis) were particularly affected as will be shown and discussed in section 3.

The CHARON-PTR-ToF-MS setup was interfaced to the SAPHIR chamber using Siltek®/Sulfinert®-treated

stainless steel tubing (total length: 600 cm, 50 cm extending into the chamber, ID: 5.33 mm). During the β-pinene

ozonolysis and limonene ozonolysis/NO$_3$ oxidation experiments, the inlet flow was kept at 0.6 l min$^{-1}$ resulting in a

sample residence time of 13.4 s. During the β-pinene/limonene mixture ozonolysis and the real tree emissions

ozonolysis experiments, the inlet flow was increased to 1.6 l min$^{-1}$ resulting in a sample residence time of 5.0 s.

**2.3.3    TD – PTR-ToF-MS**

The Thermal-Desorption unit was coupled to a commercial PTR-TOF8000 instrument (Ionicon Analytik GmbH,

Austria). The TD is a dual aerosol inlet system consisting of impact collection thermal desorption cells. The setup

was already used in several campaigns as described by Holzinger et al. (2013); (2010a).

In short, the centrepiece of both aerosol inlets is a Collection Thermal Desorption cell (CTD, Aerosol Dynamics,

Berkeley, CA, USA), on which humidified ambient particles in the size range of 70 nm to 2 μm at an air sample

flow rate of ~6 L min$^{-1}$ are collected by impaction onto a stainless steel collection surface using a sonic jet impactor.

The humidification of the aerosol sample flow to approximately 70% is achieved by a Nafion based humidifier and

reduces particle rebound. All tubing in contact with volatilized aerosol compounds (i.e. the CTD cell, and all transfer




tubing and valves) is coated to increase the chemical inertness of the surface. The CTD cell coating is AMCX
(AMCX, L.L.C., Lemont PA, USA); all other parts received the Siltek®/Sulfinert®- treatment. The transfer lines are
operated at elevated temperatures of 200 ℃ to avoid re-condensation of desorbed aerosol compounds.

In this study, aerosols were sampled from the chamber through a ~5 m long copper line (ID=6.5 mm). The operation
of the system was fully automated. One cycle was completed in 2.5 h and included the analysis of (i) the first aerosol

inlet (namely inlet A), (ii) the second aerosol inlet (namely inlet B), (iii) inlet A and (iv) inlet B that sampled
particle-filtered chamber air, and (v) the analysis of gas-phase in conventional PTR-MS mode. The duration of each

section was 30 min. Due to lab air contamination the conventional PTR-MS gas-phase measurements of the chamber
air were not available from the TD-PTR. In addition, inlet A data quality was affected by a systematic change of the

PTR-MS conditions (E/N fluctuation during background measurements caused by a malfunctioning valve).
Consequently, inlet A data were excluded from this campaign.

The aerosols were pre-concentrated onto the CTD cell for 30 min with a flow of 6 L min$^{-1}$ before thermal desorption
into the PTR-MS. After collection, a small flow of ~ 10 mL min$^{-1}$ of nitrogen carrier gas transported all compounds

desorbing from the CTD cell directly into the PTR-MS. Aerosol compounds were thermally released from the CTD-
cell by ramping the temperature up to 350 ℃ from room temperature (normally, 25 ℃). Temperature ramped

continuously at a rate of ~15 ℃ min$^{-1}$ for ~21 minutes until 350 ℃ followed by a dwell time of 3 minutes (at 350
℃). After a cool down period of 6 min a new collection was initiated. For the last experiment (tree emissions), a

denuder was installed on inlet B to constrain a possible artefact from gas-phase compounds adsorbing on the CTD
cell.

The aerosol background was measured every other run by passing the airstream through a Teflon membrane filter
(Zefluor 2.0 μm, Pall Corp.) that removed the particles from the air stream (sections: iii and iv mentioned above).

The effective removal of particles was confirmed by test measurements with a condensation particle counter (TSI,
WCPC Model 3785). While particles are removed by the Teflon filter, gas-phase compounds should be less affected.

Filter samples to determine the aerosol background have been taken by turns: in each cycle, inlet A and inlet B
sampled successively for 30 min of each, then the samples collected through the two inlets were analysed

successively as well.

       The PTRMS measures mixing ratios of compounds desorbed from aerosols in a nitrogen carrier gas. The mass

concentration of an aerosol compound in the air sample is calculated according to

$$n_{aer,x} = C_{mean,x} \times \frac{F_{N_2} \times t_{meas}}{22.4 \times F_{col} \times t_{col}},  \qquad (2)$$

where $n_{aer,x}$ is the aerosol concentration of compound X in μg m$^{-3}$, $C_{mean,x}$ its (arithmetic) mean mixing ratio during
the aerosol analysis in the nitrogen carrier gas in nmol mol$^{-1}$, $MW_x$ the molecular weight of compound X in g mol$^{-1}$,

$F_{N2}$ the flow of the carrier gas in standard liters per minute, $t_{meas}$ the duration of the aerosol measurement in minutes,
$F_{col}$ the flow rate at which the aerosols are collected in standard liters per minute, $t_{col}$ the duration of aerosol

collection in minutes and 22.4 the volume one mole of an ideal gas will occupy in liters. Mixing ratios of most
compounds were calculated according to the method described in Holzinger et al. (2010b), which involves the use of

default reaction rate constants ($3\times10^{-9}$ cm$^3$ s$^{-1}$ molecule$^{-1}$),

       Specific conditions during the campaign were as follows: E/N = $1.6\times10^{-19}$ V m$^2$ molec$^{-1}$ (i.e. 160 Townsend units) to





ensure ionization only by $H_3O^+$, temperature of the drift tube Td = 120 ℃, and a mass resolution of m/$\Delta$m ≈ 4000.

Mass spectra were obtained on a 5s time resolution. The data were processed using the PTRwid software (Holzinger,

2015). The software has several unique features including autonomous and accurate calibration of mass scale and

the export of a uniform peak list which avoids the same ion being attributed to a slight different mass within the

limits of precision. In total, 543 organic ions represented in the "unified mass list" have been obtained.
**3**     **Results and discussions**

In order to compare the different measurement techniques a time synchronization of the three data sets was

performed. All data presented in this work have been synchronized to the ACM time with a time resolution of 4

hours. The presented time is the center of the sampling interval for all experiments.

**3.1**     **Comparison of PTR-based aerosol measurement techniques to SMPS and AMS**

Comparison of the different aerosol chemical characterization techniques to the AMS and SMPS was performed by

means of linear regression (Figure 1). Since no collection efficiency (CE) was applied to the PTR-based aerosol

measurement techniques, AMS data were treated the same way throughout this work, thus no AMS CE was

enforced. SMPS organic mass concentration was calculated assuming a density of 1.4 g cm$^{-3}$ (Cross et al., 2007). A

least orthogonal distance regression linear fit function, included in the IGOR extension ODRPack95, was used for

each instrument related to SMPS data. Results suggested that the measured fraction compared to the SMPS mass

was constant for each technique throughout the campaign. Due to experimental flaws CHARON$_{100}$ introduced a

higher than usual variability of the particle enrichment in the aerodynamic lens during two experiments, the β-pinene

ozonlysis and limonene ozonolysis/NO$_3$ oxidation (Section 2.3.2). These experiments were excluded when applying

the linear fit. CHARON$_{100}$ was able to measure 80% (1σ = ± 10%) of the SMPS mass. ACM and AMS measured

51% (± 5%) and 67% (± 10%) while TD measured 27% (± 3%) of the SMPS, respectively. TD and ACM showed

the lowest slope uncertainties (≤ 5%), thus the highest stability in terms of recovery or overall detection efficiency.

CHARON$_{100}$ and AMS followed with a slope accuracy of ~ 10%, but at higher recovery rates. All instruments

showed linear fit offset values close to zero when taking into account the error of the fit (± 3σ).

For the PTR based techniques and AMS a mass recovery underestimation could be expected due to a variety of

processes from (i) the unideal CE during particle enrichment, (ii) thermal dissociation during desorption, (iii) ionic

dissociation in the ionization region and (iv) the inability to ionize the reactant/fragment. The extent to which these

processes affect the different techniques was investigated in detail in this work.

It is well known that AMS derived mass concentrations have to be corrected for CE due to particle bounce signal

loss on the vaporizer (Canagaratna et al., 2007). Fresh biogenic SOA though have a high CE (Kiendler-Scharr et al.,

2009) and reduced bouncing effect, also observed from the relatively high AMS CE in this work (~ 0.7). ACM and

TD utilize a collection surface as well and therefore introduce a CE uncertainty with the TD setup reducing even

further the bouncing effects by humidifying the particles prior to collection. CHARON is an on-line technique

avoiding the latter loss processes thus increasing the ability of the instrument to measure the mass concentration of

the compounds generated during these experiments.



During desorption, thermal dissociation of molecules could introduce two or more fragmentation products. Canagaratna et al. (2015) reported that in the AMS organics gave rise to $H_2O^+$, $CO^+$ and $CO_2^+$ signal due to surface
evaporation and thermal break down of organic molecules at vaporizer operating temperatures down to 200 °C (under vacuum conditions). Although neutral dissociation products like $H_2O$, CO and $CO_2$ could be ionized by the
AMS, their proton affinities are lower than that of $H_2O$, thus PTR techniques would no longer ionize and detect them. On the contrary, remaining smaller organic fragmentation products with proton affinities higher than $H_2O$
would still be visible to the PTR-MS. A lack of detection of certain neutral fragments formed during thermal desorption could introduce an underestimation of the total mass, oxygen and carbon concentration for the PTR based
techniques. It should be noted that decarboxylation and dehydration reactions are strongly dependent on the temperature, pressure and the heat exposure time of the molecules. CHARON was operated at the lowest
temperature of 140 °C, under a few mbars of pressure and with the lowest heat exposure time thus avoiding the latter reactions. On the contrary, ACM and TD were operated at 1 bar and up to 250 °C and 350 °C respectively with
longer heat exposure times. To further assess whether surface evaporation for ACM and TD had an additional effect on the measurements, focus was given on the experimental case studies performed by Salvador et al. (2016) using
the TD-PTR-ToF-MS. Five authentic standard substances (phthalic acid, levoglucosan, arabitol, *cis*-pinonic and glutaric acid) were utilized to examine the response of the sampling device. If the compounds would only fragment
in the PTR-ToF-MS due to ionic dissociation, then the detected fragments should have the same volatility trend as the parent compounds since both originate from the latter. During desorption of the collected samples, fragment ions
were found to represent different volatility trends compared to their parent ions (Arabitol, cis-Pinonic Acid). These thermogram differences, originating from the same substance, promoted certain amount of neutral
fragmentation/pyrolysis in the hot TD cell.

Ionic dissociation in the ionization region of the PTR-MS is strongly affected by the PTR operating conditions and

in particular the E/N applied (Section 2.3). The lower mass concentration detected by the TD unit compared to the other techniques could be partly explained by the different E/N used, with TD operated at the highest E/N = 160 Td.
This high potential of fragmentation losses during quantification would be given as:

$$(R^+)^* \rightarrow F^+ + N \tag{3}$$

where $(R^+)^*$ is the unstable protonated reactant, $F^+$ is the protonated fragment and N is the neutral product. By increasing the fragmentation potential the neutral products would increase thus lowering the total mass
concentration detected. This could also lead to an underestimation of the ACM mass concentration compared to $CHARON_{100}$ (ACM operated at 120 Td and $CHARON_{100}$ at 100 Td) and is discussed in detail in Section 3.2. It
should be noted that the mass underestimation of the ACM due to ionic and thermal dissociation could be higher than 16% (the mass difference between the ACM and AMS). This would imply that ACM CE was higher compared
to the AMS CE during this campaign, a possible result in view of the differences of vaporizer/collector geometry (Hohaus et al., 2010).
Additional comparison between the AMS and the PTR-ToF-MS based techniques was examined by determining the bulk oxygen to carbon ratio (O:C) for all instruments (Figure 2). AMS O:C values were calculated based on the
method by Canagaratna et al. (2015). All instruments followed similar trends. O:C ratios increased with



photochemistry initiation (chamber illumination) or $NO_3$ oxidation (limonene experiment/NO injection). On the

contrary, O:C values decreased when fresh BVOC was introduced into SAPHIR and additional SOA was formed

during the tree BVOCs re-emission stage (11 – 22 h after ozone injection). When compared to AMS, all PTR-ToF-

MS based techniques showed lower O:C values. Good agreement was found between the ACM and TD O:C values

(< 3% difference). $CHARON_{100}$ measured higher O:C compared to ACM and TD (ACM lower by ~ 20-35%), an

indication that during this campaign $CHARON_{100}$ was capable of detecting more oxygenated compounds. When

comparing the β-pinene and limonene experiments, $CHARON_{100}$ had increased O:C values for experiments that

incorporated β-pinene while ACM had the opposite behavior, with higher O:C during the limonene experiment. For

the tree emissions experiment the BVOC system resulted in SOA that showed increased O:C values for all

instruments introducing compounds with higher oxygen content in the particulate-phase. During this experiment

CHARON was operated at different E/N operating conditions thus providing further insights of the influence of E/N

on O:C values (Figure S2). Results showed that O:C increased by approximately 10% when changing the CHARON

E/N from 100 Td to 65 Td, thus providing softer ionization conditions.

Although nearly all $C_xH_yO_z$ ions can be identified and quantified within the AMS mass spectra, AMS O:C

calculation based on Canagaratna et al. (2015) has several sources of uncertainties due to correction factors applied.

As stated by Canagaratna et al. (2015), the overall errors observed in elemental ratios calculations would introduce

an upper uncertainty of 28%. In contrast to AMS data O:C ratios for the PTR based techniques were calculated with

no additional correction factors thus explaining their lower values when compared to AMS.

PTR-ToF-MS is considered a soft ionization technique which suffers less from fragmentation and therefor should

provide O:C ratios closer to the true values compared to uncorrected AMS data. Nevertheless, water clustering and

carbon-oxygen bond breakage could occur, either increasing or decreasing O:C ratios. When proton transfer

reactions induce fragmentation a neutral fragment is lost. For oxygenated organics it has been shown that the loss of

water as neutral fragment is a common fragmentation pathway (de Gouw and Warneke, 2007). This could explain

the lower O:C values seen from CHARON, ACM and TD compared to the AMS. Intercomparison of the PTR based

techniques further showed that $CHARON_{100}$ was more sensitive to oxygenated compounds compared to ACM and

TD. Higher O:C ratios were observed when comparing $CHARON_{65}$ to $CHARON_{100}$ indicating that low E/N values

can decrease the loss of neutral fragments such as water or carbon containing compounds with O:C ratios >1 (e.g.

$CO_2$, HCOOH). This factor does affect the ACM and TD O:C ratios even more, since they are operated at even

higher E/N (120 Td and TD at 160 Td, respectively) than CHARON. It should be noted that lower E/N values could

also increase the tendency to detect water clusters, i.e. $AH^+(H_2O)_n$, where A is the ionized organic compound,

bearing the risk to bias the O:C ratio high which is explored further in the next section.

As previously discussed, AMS $H_2O^+$, $CO^+$ and $CO_2^+$ signals are generated due to surface evaporation at

temperatures exceeding 200 °C (under vacuum conditions). These fragment signals cannot be detected from ACM

and TD (that also undergo surface evaporation compared to CHARON), thus an additional underestimation of their

O:C values could not be excluded. To assess the extent of surface fragmentation, further re-calculation of the AMS

O:C, excluding the $H_2O^+$, $CO^+$ and $CO_2^+$ peaks (Figure S3) was performed and compared to the PTR-based

techniques. Results showed that AMS O:C ratios was lower than that of ACM and TD. When only excluding the



$H_2O^+$ signal, AMS O:C ratios were higher than those of ACM and TD. These results suggest that CO and $CO_2$ loss

by thermal dissociation in the ACM and TD play a less significant role compared to AMS due to their lower

operating evaporation temperatures and higher pressure.

When comparing experiments incorporating β-pinene or limonene, the different behavior of the O:C ratios found for

the $CHARON_{100}$ (O:C $_{CHARON, limonene}$ < O:C $_{CHARON, β-pinene}$) and ACM (O:C $_{ACM, limonene}$ > O:C $_{ACM, β-pinene}$) could be

due to different fragmentation patterns of the particulate-phase functional groups or due to their volatility

differences. Since limonene SOA are less volatile than β-pinene SOA (Lee et al., 2011) a fraction of the OA

oxygenated mass that would evaporate at higher temperatures could be lost, thus leading to lower O:C values

compared to the β-pinene experiments. However, ACM showed only minor volatility differences when comparing

the β-pinene to the limonene experiments, as seen in Figure S4. Although CHARON was operated at lower

temperatures compared to ACM, its reduced pressure compensated for the temperature difference thus increasing the

volatility range down to LVOC (Eichler et al., 2017). These results conclude that differences in the O:C trends of

ACM and CHARON could not be explained by changes of the SOA volatility. The ionic and thermal dissociation

patterns of the different particulate-phase functional groups could play a role in these findings and has to be

examined in future studies.

### 3.2     Classification of SOA composition

Further comparison of the aerosol chemical characterization techniques was performed with a focus on the different

chemical characteristics (oxygen content, carbon content, molecular weight) of the SOA composition. A desorption

period from the tree emissions experiment, 25 hours after the ozone injection (Figure 2 (d)), was chosen in order to

highlight the instrument performance differences, shown in Figure 3. The mass concentration of all compounds

containing the same carbon number was calculated. These carbon fractions were then further separated depending

on the number of oxygen atoms the compounds contained. The molecular weights (MW) of the SOA constituents

was separated in five different *m/z* range groups, from *m/z* 30 - 50, *m/z* 50 - 100, *m/z* 100 - 150, *m/z* 150 - 250, *m/z*

>250. All instruments showed similar carbon content distributions, with the highest concentration introduced from

C8 compounds. CHARON was able to measure compounds in the C10 - C20 range while ACM and TD only

detected up to C13 compounds. The overall OA mass concentration decreased when moving from lower

($CHARON_{65}$ and $CHARON_{100}$) to higher E/N values (ACM at 120 Td and TD at 160 Td). The same trend was seen

for the oxygen content of compounds; with a characteristic example being the compounds containing 5 oxygen

atoms that decreased by a factor of 2 with the same instrument but different operational parameters for the PTR-

ToF-MS ($CHARON_{65}$ vs. $CHARON_{100}$). In ACM and TD compounds containing 5 oxygens were negligible. A

similar trend was observed for *m/z* range distributions, with a higher fraction of low *m/z* compounds observed at

increasing E/N values. ACM and TD results indicated that the main fraction of compounds was detected for MW <

100 amu (70 and 75% of the overall mass concentration, respectively).

These results clearly show the high dependency of the overall mass concentration detection as well as the carbon,

oxygen and MW content determination being strongly affected by the PTR-ToF-MS E/N operating conditions. As

the E/N values increased, oxygen-carbon bond breakage increased leading to undetected neutral fragments. This loss





of information directly affects the overall mass concentration and MW detection range. Comparing the ACM to the TD MW pie charts showed that, although ACM was operated at lower E/N conditions (120 Td) than the TD (160
Td) the contribution in the lower MW range was higher for the ACM. The reason for this dissimilarity could be due to the lower resolution and the higher limit of detection of the PTR-ToF-MS used for the ACM (see Table 2) leading
to lower detection of the higher molecular weight compounds. Since water loss is the major fragmentation occurring in the PTR-ToF-MS, the oxygen content is affected the strongest. This could explain why compounds with 5
oxygens were nearly undetectable for ACM and TD compared to CHARON.

To further assess the differences in chemical classification by each instrument the relative OA mass concentration of

molecular carbon, oxygen and weight (box-and-whiskers including all data points throughout the campaign) were used, as seen in Figure 4. ACM and TD showed similar distributions for all contributions throughout the campaign
with only minor differences (< 3%). On the contrary, their comparison to CHARON$_{100}$ showed a clear difference. Compounds in the lower MW range (< $m/z$ 150), containing lower molecular carbon (< 9 carbon atoms) and oxygen
(< 2 oxygen atoms) showed higher contributions for the ACM and TD compared to CHARON$_{100}$. A detailed comparison of CHARON's different E/N conditions during the tree emissions experiment (Figure S5) was also
performed. Results indicated that for lower E/N, an absolute difference of 2%, 5% and 10% for the molecular carbon, weight and oxygen contributions were observed, respectively, suggesting that in this E/N range (from 65 to
100 Td) fragmentation is dominated by oxygen containing functional groups loss.

The above results strongly suggest that the E/N settings play a key role to the fragmentation patterns. By increasing

the drift tube voltage, the velocity of the ions increased, leading to higher kinetic energy in ion molecule and therefore stronger buffer gas collision. This energy increase was translated to an increase in carbon-oxygen bond
breakage. On the contrary, the lower the E/N was set, the higher the sensitivity due to enhanced reaction times but also the stronger the cluster ion distribution change, supporting more $H_3O^+(H_2O)_n$ (n=1,2,3) cluster ion generation
(de Gouw and Warneke, 2007). In order to quantify whether the PTR-ToF-MS E/N conditions were a major factor for the differences seen during this campaign, a case study of pinonic acid was performed in the lab. Monodisperse
pinonic acid particles were generated (900 – 1100 particles/cm$^3$) and directed to a CHARON-PTR-ToF-MS, changing E/N values from 60 to 170 Td (Figure S6). Results showed that the relative intensity of the parent ion
decreased rapidly when increasing the E/N values. At the same time, the relative intensity of the lightweight fragments was increasing. The effect of the parent ion clustering with water was negligible suggesting no
overestimation of the CHARON oxygen content at low E/N (65 Td). By assuming a uniform sensitivity and calculating the total signal (parent ion and fragments, assuming all $m/z$ represent parent molecules) the mass fraction
of pinonic acid particles was calculated (Figure S7). The higher the E/N values were set, the less the PTR-ToF-MS measured compared to the SMPS. These results confirmed our previous findings that fragmentation losses lead to an
underestimation of the overall mass concentration. Therefore the different E/N conditions of the detection systems (PTR-ToF-MS) could explain in a large fraction the differences between the CHARON, ACM and TD oxygen and
carbon content (results seen in Figure 2 and Figure 4) as well as their differences in the overall detectable mass (results seen in Figure 1 and Figure 3). A clear influence of the aerosol sampling technique on the differences of
these parameters cannot be determined nor excluded (Salvador et al., 2016).





### 3.3 Volatility comparison

During the campaign, CHARON was operated at a constant temperature (140 °C) while ACM and TD ramped through different temperatures during desorption of the collected aerosol samples (see Section 2). The ramping of
ACM and TD provided the possibility of a detailed comparison of the compound dependent volatility trends. In Figure 5 the timeseries of ACM and TD for the β-pinene, the β-pinene/limonene mixture and the tree emissions
experiments were investigated. The limonene ozonolysis and $NO_3$ oxidation was excluded from this comparison, due to TD operational problems. For both instruments high contributions of the aerosol mass concentration
evaporated at lower temperatures when fresh SOA were generated (initial hours of the experiments and tree emissions $A_o$ stage), hence higher SOA volatility values were observed. As oxidation continued the relative
contributions of aerosol mass evaporating at low temperatures and therefore the overall volatility decreased. When illuminating the chamber, SOA volatility decreased suggesting that photochemical aging of the SOA took place
leading to a change of the chemical composition and volatility distribution. For experiments having β-pinene as a precursor for the subsequent SOA formation, TD showed a decreasing volatility as the experiment evolved while
ACM reached a plateau after 5 to 10 hours of aging.

The volatility changes for both instruments, during the initial hours of the experiments and during the re-

introduction of BVOCs for the trees experiment, could be attributed to the high concentration semi-volatile organic compounds (SVOCs) in the gas-phase that had the maximum available surface to condense on (SMPS at its
maximum surface area and mass concentration). Under these conditions, these compounds would partition more to the particulate-phase thus increasing their contribution during the highest concentration periods. These easier to
evaporate SVOCs could change the volatility patterns as observed from both techniques by a change of the thermograms during the maximum concentration periods. Discrepancies between the ACM and TD, with the latter
having a steadily changing desorption temperature with time, could be affected by several operating differences. During evaporation ACM was ramped by 100 °C min$^{-1}$ to a maximum of 250 °C, with 3-minute isothermal sections
at 100 °C, 150 °C and 250 °C, respectively, while TD was ramped continuously at a rate of ~15 ℃ min$^{-1}$ for ~21 minutes until 350 ℃. The higher volatility resolution of TD compared to ACM could introduce an increased
sensitivity to volatility changes thus increase the TD variability compared to ACM. Differences could be partly attributed to the different design of the instruments. ACM ensured complete separation of the particulate from the
gas-phase (> 99.9999 gas-phase removal) while TD was corrected for gas-phase contamination by performing background measurements (Section 2). As the collection of the particulate-phase compounds was performed for the
TD, the collector was exposed to high concentration of SVOCs from the gas-phase, thus increasing the absorption of these compounds to the particulate-phase. As the gas-phase concentrations decreased the TD volatility decreased.
This could thus indicate a possible background correction artifact mostly affecting compounds in the higher volatility range, evaporating in the first temperature steps (100 ℃). It should be noted that after the β-pinene initial
hours of consumption, secondary reactions in the absence of light and the presence of ozone should be negligible due to the lack of unsaturated reactants. The expected temporal volatility behavior would thus be shifted towards a
more stable instead of changing volatility system.





To further assess the volatility differences of ACM and TD, focus was given on the molecular oxygen number based on the assumption that oxygen number correlates to volatility (Jimenez et al., 2009). Box-and-whiskers, including all campaign desorption periods, were generated for each molecular oxygen number at each temperature, as seen in Figure 6. The data were normalized to the sum of the measured mass concentration from each molecular oxygen number in all temperatures (top equation in Figure 6). Results showed that TD had a broader range in fractional contribution for all oxygen bins when compared to the ACM. A characteristic temperature showing this difference was at 150 °C, where TD showed results in the range of 0.2 to 0.55 while ACM was in the range from 0.15 to 0.25. Despite the differences in relative contribution, both instruments showed similar trends. As the collector temperature increased oxygenated compounds (2, 3 and 4 oxygens) contributed more than lower oxygenates. On the contrary, at lower temperatures compounds containing 0 and 1 oxygen were the dominant factor. Overall, for ACM around 20% of the SOA evaporated at 100 °C, 20% at 150 °C and 60% at 250 °C. TD showed similar volatility trends with 15 to 20% of the SOA evaporating at 100 °C, 35% at 150 °C and 50 to 55% at 250 °C.

According to observations and theory (Jimenez et al., 2009) oxygenated compounds are expected to have lower volatility thus evaporating at higher temperatures. TD and ACM described the expected volatility trends during the performed experiments based on compound specific information in accordance to theory. The variability of TD compared to ACM reflected the differences in the design and operation of the individual systems described previously. The higher volatility resolution but also the higher E/N conditions of TD could explain most of the observed discrepancies. Fragmentation due to ionic dissociation after the evaporation could influence the volatility molecular oxygen content distribution by loss of neutral oxygen containing fragments. This could further affect the volatility distribution when the oxidation product concentrations change with time, reflected by the increase of the O:C ratios (see Figure 2). Furthermore, the ability of ACM to achieve complete gas to particle separation resulted in a lower thermogram uncertainty in the higher volatility range thus smaller variations. These results show the applicability of both techniques to study BSOA volatility trends in a compound specific level.

### 3.4    Compound detection comparison and tracers attribution

The molecular formula ($C_xH_yO_zN_a$) was attributed to each detected signal derived from the exact molecular mass (see Section 2) determined by the TOF-MS for all 3 techniques throughout the campaign. In order to assess whether major contributing molecules with the same chemical characteristics were determined by all instruments, a comparison of the dominant signals was performed i.e. the molecular formulas that (i) were measured by all techniques during each experiment and (ii) were within the 80 highest signal concentrations. Figure 7 shows the respective results from the BSOA detected in the C7 to C10 range with varying oxygen content (from 0 to 4 oxygens). Although these techniques could provide the molecular formula of the compounds, the molecular structures are unknown. In order to derive further information, comparison to previous publications was performed for the major oxidation products from (a) the β-pinene ozonolysis (Chen and Griffin, 2005; Hohaus et al., 2015; Jenkin, 2004; Yu et al., 1999), (b) limonene ozonolysis and $NO_3$ oxidation (Chen and Griffin, 2005; Jaoui et al., 2006; Kundu et al., 2012; Leungsakul et al., 2005a; Leungsakul et al., 2005b) and (c) tree emissions ozonolysis with α-pinene and $\Delta^3$-carene being the major reactants (Chen and Griffin, 2005; Praplan et al., 2014; Yu et al., 1999).



Results showed that all techniques were able to detect most of the expected molecules. Details on the molecular formula and suggested structure are provided in more detail in Table S1. Due to fragmentation most of the
compounds were not detected at the parent ion molecular weight but underwent water loss in accordance to the findings that O:C ratios are observed to be reduced by ACM, TD and CHARON compared to the AMS (see Section
3.1). These compounds corresponded to a large fraction of the BSOA mass measured from each technique (bars in Figure 7). On average,70%, 60%, and 40% of the measured mass was contributed from these compounds, for ACM,
CHARON and TD respectively. When comparing the above compounds concentration to the SMPS total mass, around 30%, 50% and 10% of the SMPS mass for ACM, CHARON and TD respectively was explained. The
overlapping of detected compounds to previous publications (theoretical and experimental work) and their high contribution (up to 50%) to the overall BSOA mass concentration strongly promotes the use of PTR-ToF-MS aerosol measurement techniques to gain valuable insight on the chemical characteristics of BSOA.
**4     Conclusions**

        A comparison of three different aerosol chemical characterization techniques has been performed as part of a

chamber study on fresh and photochemically aged BSOA, formed from the ozonolysis of monoterpenes. The aerosol collection module (ACM), the chemical analysis of aerosol on-line (CHARON) and the collection thermal
desorption unit (TD) are different aerosol sampling inlets utilizing a PTR-ToF-MS. These techniques were deployed in a set of chamber experiments at the atmosphere simulation chamber SAPHIR to investigate SOA formation and
aging from different monoterpenes ($\beta$-pinene, limonene) and from real plant emissions (Pinus sylvestris *L.*).

        The total aerosol concentration recovery of the PTR based techniques, compared to an SMPS, was

$80 \pm 10\%$, $51 \pm 5\%$ and $27 \pm 3\%$ for CHARON, ACM and TD respectively. In contrast, an AMS concurrently operated and with no collection efficiency correction applied, showed a recovery of 67%. The three PTR based
techniques were capable of measuring the same major contributing signals for the different monoterpene oxidation products studied. These attributed compounds corresponded to a high fraction of the overall SOA mass
concentration with 30%, 50% and 10% of the overall mass being explained for ACM, CHARON and TD respectively. Additional comparison to previous publications showed that these compounds corresponded to known
products of the monoterpenes studied. Both the ACM and TD collection and thermal desorption design provided additional information on their volatility and showed similar trends. Compounds containing higher molecular
oxygen number ($\geq 2$) contributed more to the aerosol fraction desorbed at high temperatures than lower oxygenated compounds (molecular oxygen number < 2) which were more efficiently desorbed at low temperatures.
Oxygen to carbon ratios (O:C) increased while SOA production and ageing proceeded. All instruments had comparable O:C trends during the course of an experiment. Good agreement was found for the ACM and TD O:C
values (< 3% difference) while CHARON showed 20 to 35% higher O:C ratios.

        Despite significant difference in the aerosol collection and desorption techniques, the major reason for the

discrepancies was the different operating conditions of the PTR-ToF-MS. Laboratory case studies supported that E/N conditions played a crucial role in carbon-oxygen bond breakage leading to lower O:C ratios at high E/N. Since
ACM and TD were operated at higher E/N compared to CHARON this resulted to higher fragmentation, thus affecting their oxygen and carbon content and mass recovery. Compared to AMS, PTRMS is a soft ionization





technique even at high E/N and therefore less prone to fragmentation. AMS requires correction factors (Canagaratna et al., 2015), to determine O:C ratios wereas for PTRMS corrections were omitted. Determination of O:C ratios for
the PTR based techniques was thus underestimated, explaining their difference to the HR-ToF-AMS (30 to 50% higher). Differences in the sampling and evaporation technique might introduce also deviations between the
chemical characterizations i.e. due to thermal decomposition. This has to be studied in detail in future comparisons by operating the PTR-ToF-MS instruments under the same E/N conditions.
The ability of all PTR based techniques to measure compounds, supported from previous publications, strongly promotes their use. These techniques can provide valuable insight on the chemical characteristics of freshly formed
and aged BSOA, and on thermodynamic properties such as partitioning coefficient values and volatility patterns on a compound specific level.

**Author contribution**

RT , RH, AW and AKS designed the experiments. TH and RT operated the chambers. SHS, PS, PE, MM, KM, GIG, RW, MK, AW, RH and RT conducted the data collection and evaluation for AMS, TD, CHARON, ACM, PTR and
GC-MS. MM designed and carried out the laboratory characterization experiments. GIG, RT, TH and AKS did the data analysis. GIG did the data interpretation and prepared the manuscript with contributions from all co-authors.

**Acknowledgements**

This work is supported by the EC's 7th Framework Program under Grant Agreement Number 287382 (Marie Curie Training Network PIMMS), by the Helmholtz President's Fund (Backfeed), and by the Dutch NOW Earth and Life
Science (ALW), project 824.14.002.The authors acknowledge support by the SAPHIR teams, electronic and mechanical workshops.

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





**Table 1: Experimental conditions for each experiment. For the tree emissions experiment there were two VOC injection periods.**


| Experiment | Monoterpenes (ppb) | Ozone (ppb) | Duration (h) | Maximum SOA formed ($\mu g/m^3$) | SOA formation conditions | SOA aging Conditions |
|---|---|---|---|---|---|---|
| **β-Pinene** | 120 | 700 | 34 | 130 | Ozonolysis | Photochemical oxidation for 10 h |
| **Limonene** | 25 | 150 | 17 | 50 | Ozonolysis | Continuous $NO_3$ oxidation for 8 h |
| **β-Pinene/Limonene mixture** | 60/12 | 300 | 26 | 60 | Ozonolysis | Photochemical oxidation for 4 h |
| **Tree emissions** 1st inj. / 2nd inj. | 65/10 | 300 | 30 | 80 | Ozonolysis | Photochemical oxidation for 6 h |
















**Table 2: Instruments operating conditions.**

| INSTRUMENT CHARACTERISTICS | ACM (in situ) | CHARON (online) | TD (in situ) |
|---|---|---|---|
| **Time resolution (min)** | 240 | 1 | 120 |
| **Gas/particle separation** | High vacuum | Denuder | Denuder and/or blank correction (filtered air) |
| **Pre-concentration factor** | 3 | 44 | 10000[a] |
| **LOD[b] (ng/m³)** | 250[c] | 1.4[d] | 0.001[a] |
| **Temperature range (°C)** | 25 – 250 | 140 | 25 – 350 |
| **Heating rate (°C / min)** | 100 | 0 | 15 |
| **Temperature steps (°C)** | 100, 150, 250 (3 min) | none | None |
| **Desorption pressure (atm)** | 1 | < 1 | 1 |
| **Particle range (nm)** | 70 – 1000 | 70 – 1000 | 70 - 2000 |
| **PTR-ToF-MS E/N (Td)** | 120 | 65 / 100 | 160 |
| **PTR-ToF-MS mass resolution (m/Δm)** | 2500 | 4500-5000 | 4000 |

[a] based on 30 min sampling at 9 L/min and 3 min desorption at 9 mL/min (Holzinger et al., 2010a)

[b] Limit of detection

[c] For signal on $m/z$ 139 and 10 sec integration time

[d] For signals around $m/z$ 200 and 1 min integration time









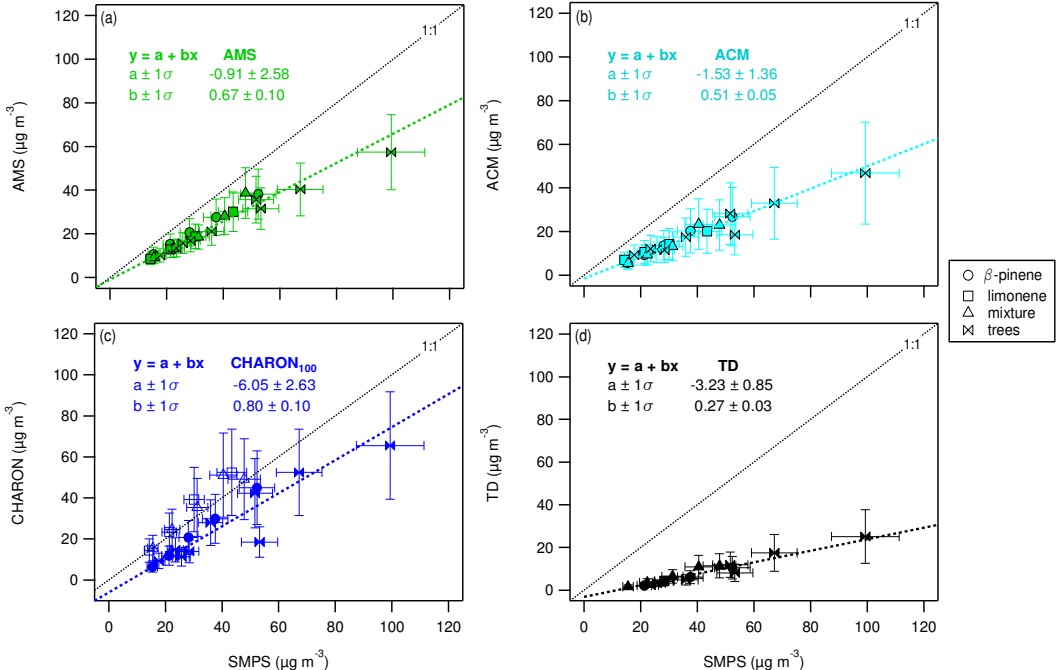

**Figure 1: Comparison of the organic mass concentration of (a) AMS (green), (b) ACM (ciel), (c) CHARON₁₀₀ (blue) and**
**(d) TD (black), to the SMPS (x-axis). Markers correspond to the different experiments with the mixture experiment**
**accounting for the mixture of β-pinene and limonene. AMS data presented are not corrected for collection efficiency.**
**CHARON₁₀₀ corresponds to data taken only at 100 Td E/N operating condition. Error bars provide the uncertainty of**
**each instrument (details in Section 2.3). A least orthogonal distance regression linear fit is applied for every instrument,**
**taking into account all campaign measurement points. Exception is the CHARON limonene and mixture data (unfilled**
**markers) that were excluded due to experimental flaws. Details of the coefficient values and their standard deviation are**
**given on the upper left of each graph.**














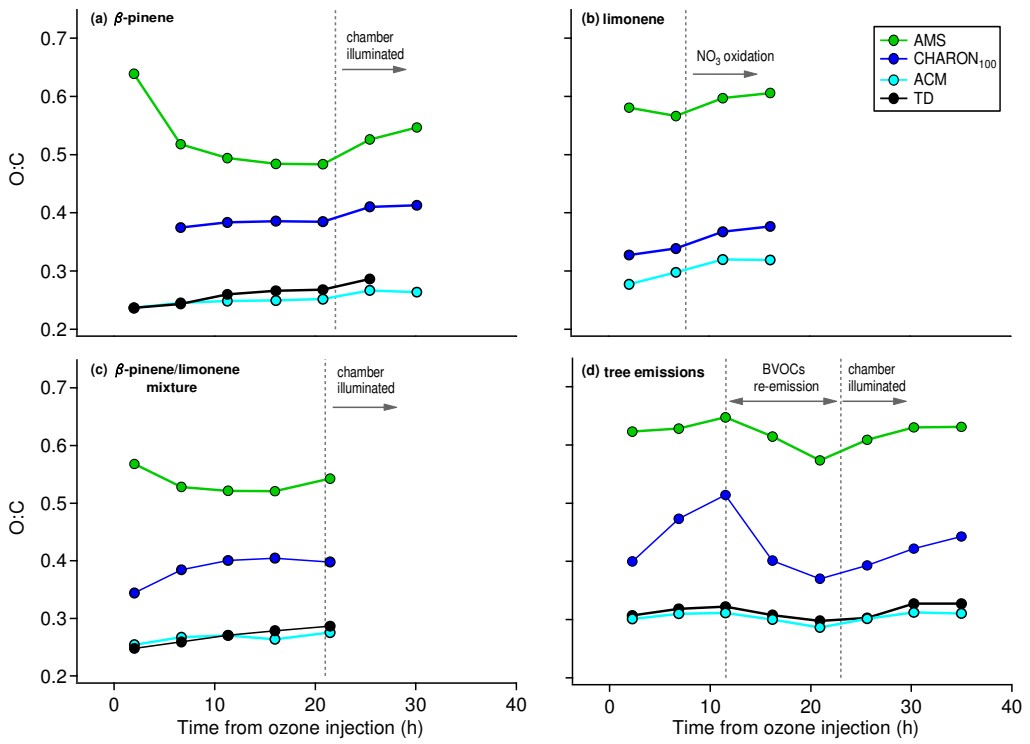

**Figure 2: Bulk oxygen to carbon ratio comparison for the different instruments (CHARON$_{100}$: blue, AMS: green, ACM: ciel, TD: black) versus the time from ozone injection. Experimental description details are provided in Table 1.**





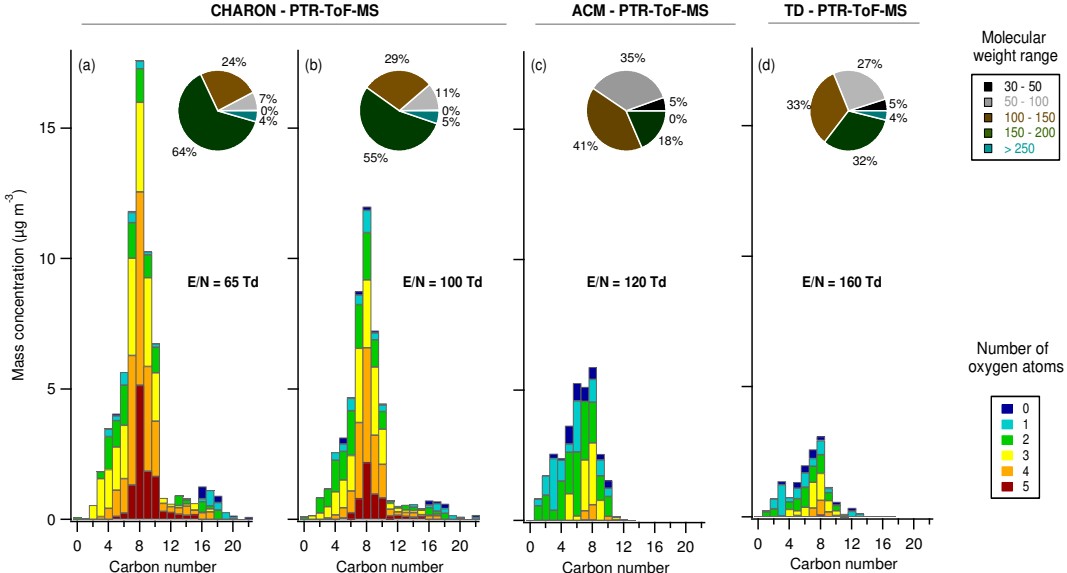

**Figure 3: OA mass concentration (y-axis) distributed based on the number of carbon atoms (x-axis). Bar colours correspond to the contribution of oxygen atoms starting from 0 (blue) to 5 (red) for each carbon group when (a) CHARON was operated at E/N = 65 Td, (b) CHARON operated at 100 Td, (c) ACM operated at 120 Td and (d) TD operated at 160 Td. Pie charts correspond to the molecular weight contribution to the overall mass starting from *m/z* 30 – 50 (black) up to *m/z* > 250 (ciel). Results shown in this graph are from the tree emissions experiment at a high OA mass concentration, 25 h after the ozone injection (Figure 2 (d)).**





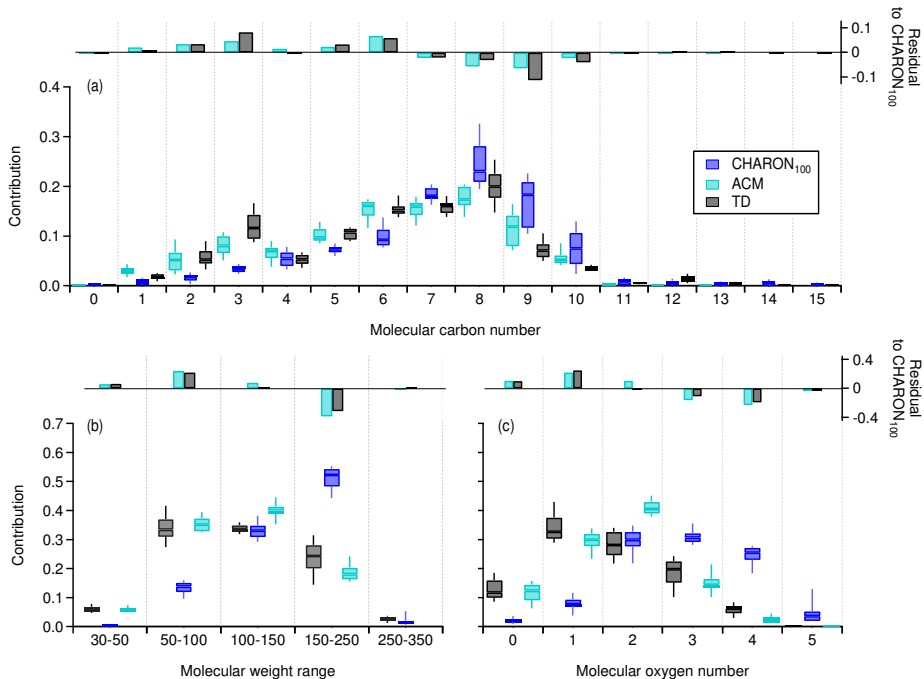

**Figure 4: Box-and-whisker plots showing the relative OA mass concentration distribution dependent on (a) molecular carbon number, (b) molecular weight and (c) molecular oxygen number for the different instruments, indicated with**
**different colours (CHARON[100] blue, ACM ciel and TD black). Each box-and-whisker corresponds to the median, 25th and 75th percentile levels of all data throughout the campaign. Upper graphs indicate the difference between the ACM and TD**
**to the CHARON[100] median values defined as residual to CHARON[100].**











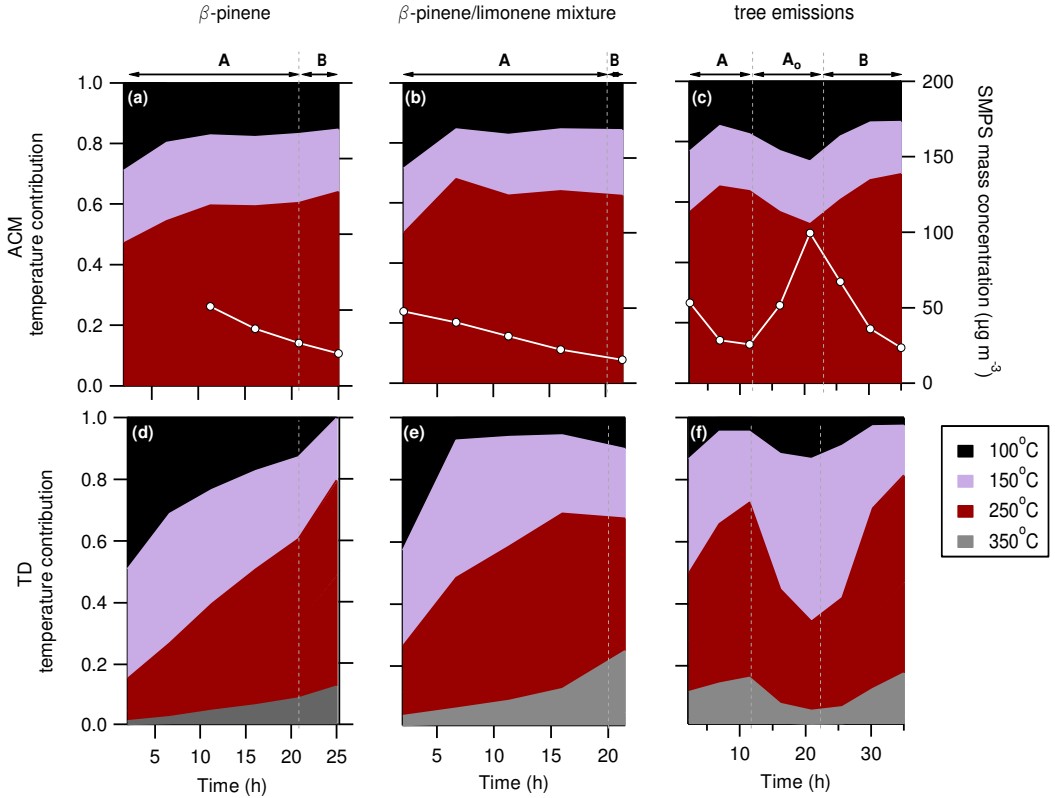

**Figure 5: Temperature dependent mass concentration contribution (left y-axis) of ACM (upper plots: a, b, c) and TD (lower plots: d, e, f) for β-pinene (a, d), β-pinene and limonene mixture (b, e) and real tree emissions (c, f) versus the time since ozone injection (x-axis). White lines and circle markers (right y-axis) represent the SMPS mass concentration during each experiment. Dash vertical lines indicate the different experimental periods with A: the ozonolysis and SOA formation period, B: the chamber illumination and photo-oxidation period and A₀: the tree emissions BVOCs re-injection to the SAPHIR chamber.**



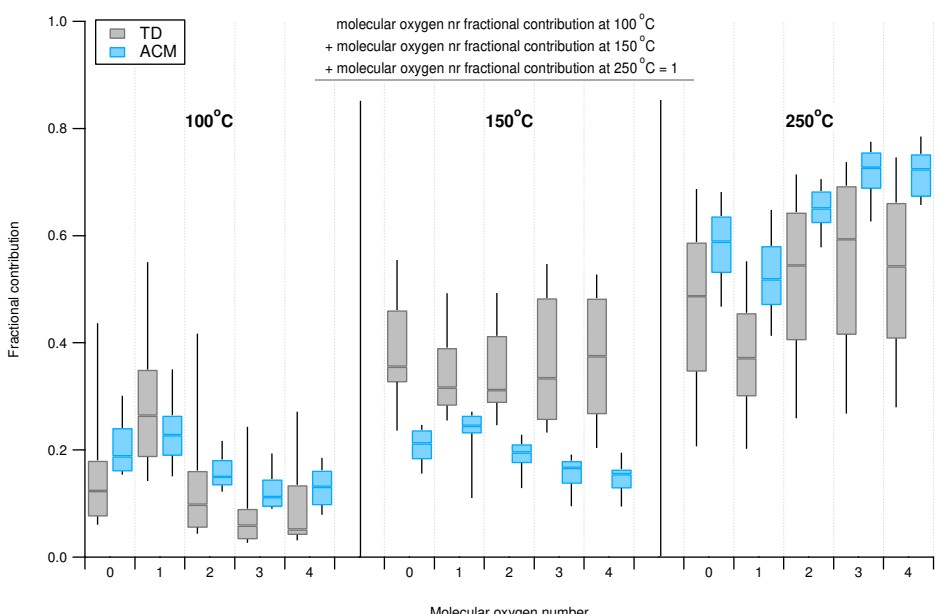


**Figure 6: Box-and-whisker plots showing the distribution of the molecular oxygen number (x-axis), for the different**
**temperature steps (100 °C, 150 ºC, 250 °C) of ACM (ciel) and TD (black). Each box-and-whisker corresponds to the**
**median, 25$^{th}$ and 75$^{th}$ percentile levels of all desorption points throughout the campaign. Upper equation indicates how the**
**contribution of each molecular oxygen number, at each temperature, corresponds to unity.**











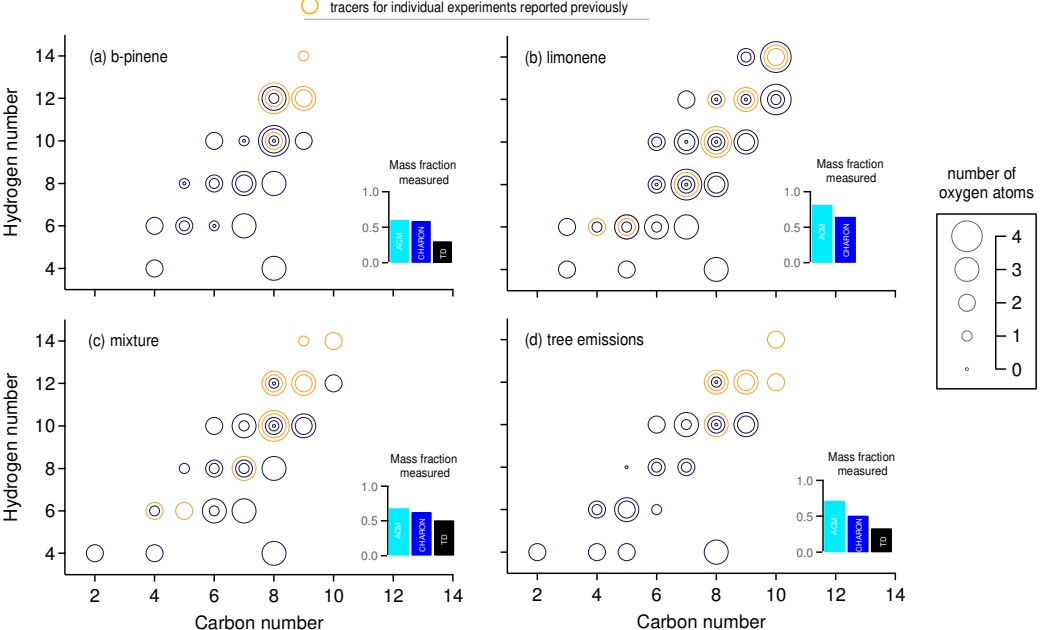


**Figure 7: Chemical formula attribution based on the molecular carbon number (x-axis), hydrogen number (y-axis) and oxygen number (markers size) for (a) the ozonolysis of β-pinene, (b) ozonolysis and NO₃ oxidation of limonene, (c) ozonolysis of the β-pinene/limonene mixture and (d) ozonolysis of real tree emissions (Scotts pine). Markers correspond to compounds measured from all techniques (ACM, CHARON and TD) at high concentrations (within the 80 compounds observing highest concentration). Each circle corresponds to one compound. Orange markers indicate tracer compounds supported from previous publications (for details refer to Table S1). Bars indicate the fraction of mass explained when accounting only the presented compounds, for each instrument (ACM ciel, CHARON₁₀₀ blue and TD black) based on their total aerosol mass measured.**