# Peer review of "Comparison of three aerosol chemical characterization 2 techniques utilizing PTR-ToF-MS: A study on freshly formed and aged biogenic SOA"

_Atmospheric Measurement Techniques, 2017_

## Referee Comment (RC1) · Anonymous Referee #1 · 17 Oct 2017

The paper compares organic aerosol composition measurements made by three different instruments at the SAPHIR chamber. The instruments are each based on proton-transfer-reaction time-of-flight mass spectrometry, but differ in the way that aerosol is sampled, evaporated and injected into the drift tube. The work is insightful and deserves to be published after consideration of the following major and detailed comments:

A. There is virtually no discussion of nitrogen-containing ions in the measurements. Did these only constitute a minor fraction of the total signal? In PTR-MS, nitrate species

commonly fragment into a nitric acid neutral and hydrocarbon ion. To what extent is that fragmentation channel responsible for some of the incomplete detection of mass shown in Figure 1? As an aside, it is difficult to appreciate how much of the data shown in the various graphs was taken under no-NOx conditions vs. conditions with NOx present.

B. The quoted detection limits differ by 5 orders of magnitude between the three instruments. To what extent can these differences be understood in terms of the sampled mass, dilution flows, sensitivities and time responses for the different PTR-TOF-MS systems used?

C. The paper describes the aerosol sampling used in the three instruments in great detail, which is appropriate. However, there is very little detail about the PTR-TOF-MS systems used. What were the types of instruments used, why is the mass resolution so different between the three systems (Table 2) and how did the primary ion signals and calibration factors compare between the three systems?

Detailed comments:

Line 34: "predominantly" instead of "predominately"

Line 37: "carbon-oxygen bond breakage" appears to be used here and throughout the text as synonymous with process that lower the O:C ratio. However, carbon-oxygen bonds are not necessarily broken in all fragmentation processes, so I would recommend the more general "fragmentation".

Line 88: "low-volatility VOCs" instead of "low VOCs"?

Table 2: Please add the temperature and pressure of the drift tube reactors used in these experiments. Also useful would be more details on the specific TOF-MS systems used and how these translate into the primary ion signals and sensitivities (in raw and/or normalized counts per seconds) of the three systems.

Lines 165-168: Limits of detection vary by orders of magnitude between the three

instruments. Part of the difference (between TD and CHARON) must be related to the time response of the methods? How do the detection limits compare if the same averaging times are used? The scatter in Figure 1 appears to indicate that the precision of the measurements is similar when averaged over the same time, but perhaps the data should not be interpreted like that.

Line 168: "V/cm" instead of "V cm"

Lines 168-169: A graph showing the different distributions of primary ions in the three different instruments would be helpful.

Line 183: The mass resolution is quoted as 2500 in Table 2. Why is there such a large difference with the specifications of the mass spectrometer?

Figure 1b: the error bars and line fitted through the data points are hard to see.

Line 348: "variability" instead of "uncertainties"?

Lines 351-354: how about incomplete evaporation of the sampled OA? The upper temperature in the three instruments is quite different and some, like the CHARON, appear to be low compared to other thermal desorption measurements. For example, Figure S1 shows that an appreciable amount of OA evaporates above 150 C. How about transmission losses of OA vapors from the evaporation zone to the drift tube?

Lines 417-429: by assuming that the AMS gives the correct elemental composition of the OA (a big if), is it possible to derive stochastically what the average composition of the undetected fragments in the CHARON measurements is?

Figure 4: these graphs might be a little easier to look at, if the ACM and TD results were shown next to each other, instead of having the CHARON results in between.

---

## Referee Comment (RC2) · Anonymous Referee #2 · 22 Oct 2017

"Comparison of three aerosol chemical characterization techniques utilizing PTR-ToF-MS: A study on freshly formed and aged biogenic SOA" by Gkatzelis et al. Atmospheric Measurements Techniques Discussions

PTR-ToF-MS provides real-time, robust measurements of ambient VOCs. This manuscript expands the use of PTR-ToF-MS applications to include particulate bound organics and compares performances of three different aerosol sampling techniques, aerosol collection module (ACM), the chemical analysis of aerosol online (CHARON), and the thermal desorption (TD) to evaluate their ability to provide chemical details of

organic aerosol when coupled with PTR-ToF-MS. It also examines their ability to provide additional information relevant to the organic aerosol such as aging, O:C ratios, and volatility patterns. The authors performed carefully designed experiments to replicate the formation and aging of biogenic SOA and did careful analyses and interpretation of the results considering different factors that can affect the experimental results as E/N. Considering the importance of organic aerosols in the atmosphere and the difficulties associated with the chemical characterization of organic aerosols, this work is valuable as it expands and improves the atmospheric measurements techniques for organic aerosol speciation. Therefore, I recommend this work for publication in AMTD after minor revisions. 1) Although this work aimed to compare performances of different aerosol sampling technique, the operating conditions and PTR-ToF-MS setups were not the same for three aerosol samplers, which affected the measured collected efficiency. The authors discusses the effect of E/N on the ionic fragmentation in the drift tube at the end of this discussion. I would move this fact to the beginning of the discussion in section 3 so there is no suspense and modify the figure captions to include the different operating conditions. 2) The authors compare the organic mass concentration corresponding to different aerosol samplers and AMS to that of SMPS. These instruments measure particles with different size ranges. There is no discussion of aerosol size distribution. It is worth to include a short discussion on measured particle size distribution and samplers' size range. Also it is not certain why the authors compared the organic mass measurements by the three samplers to that of SMPS, which is derived using a density correction. Would not this be more reliable to compare those to AMS derived mass concentration? Also this comparison implies that the aerosol volume measured by SMPS is 100% OC. Is it correct assumption? 3) As the organic aerosol age, more volatile smaller chain oxygenates can gas off the aerosol surface, thus affecting the O:C ratio and volatility. The authors discussed effect of PTR measurement conditions on the fragmentation, but not much about the fragmentation/ gassing off due to oxidation of OA. Can the authors comment or include a discussion? 4) Although the manuscript is structured well, the language and writing could be improved. It is

recommended the authors do a thorough proofreading and improve the fluency. Few examples include: Page 5 line 147: replace "where" with "was" Page 12 line 435: . . . ratios were lower that . . . instead of ". . . . . .ratios was lower that. . . . . . . . . . . . ." Page 15 line 527-530: consider re-phrasing. Page 17 line 610: "aging" instead of "ageing"

---

## Author Comment (AC1) · 15 Dec 2017

**Response to Referee #1**

We thank the referee for the helpful and constructive comments. We carefully addressed all of them as described below.

(1) The paper compares organic aerosol composition measurements made by three different instruments at the SAPHIR chamber. The instruments are each based on proton-transfer-reaction time-of-flight mass spectrometry, but differ in the way that aerosol is sampled, evaporated and injected into the drift tube. The work is insightful and deserves to be published after consideration of the following major and detailed comments:

**Major comments:**

(2) There is virtually no discussion of nitrogen-containing ions in the measurements. Did these only constitute a minor fraction of the total signal? In PTR-MS, nitrate species commonly fragment into a nitric acid neutral and hydrocarbon ion. To what extent is that fragmentation channel responsible for some of the incomplete detection of mass shown in Figure 1? As an aside, it is difficult to appreciate how much of the data shown in the various graphs was taken under no-NOx conditions vs. conditions with NOx present.

This is a very good comment, as fragmentation of nitrogen-containing ions can indeed lead to an underestimation of the total signal measured by PTR-MS. However, all experiments except the limonene SOA aging were conducted under low-NOx conditions. The maximum number of data points for each instrument being related to high-NOx conditions in Figure 1 is two, which is in terms of particle mass recovery insignificant. Further insights on the organic nitrate fraction of the SOA mass have been gained by AMS measurements. To emphasize that organic nitrates constitute a minor fraction of the particulate phase, the respective AMS results are provided in the supplement together with the gas-phase NOx measurements. We therefore pick up the discussion of potential influence of nitrogen-containing compounds in section 3.1 by inserting 'Commonly occurring neutral fragments are $H_2O$ from organic hydroxyl functional groups or $HNO_3$ from organic nitrate functional groups. While the former is often observed, during our studies organic nitrate fragmentation has not been observed as their formation is hindered during our experiments due to low NOx-conditions. This has been supported by AMS derived organic nitrate measurements being below 10% (Figure S4).'

(3) The quoted detection limits differ by 5 orders of magnitude between the three instruments. To what extent can these differences be understood in terms of the sampled mass, dilution flows, sensitivities and time responses for the different PTR-TOF-MS systems used?

Limits of detection between the three PTR-based techniques strongly differ due to the different pre-concentration factors and integration times used. The values provided in Table 2 therefore reflect the sampling and detection aspects limiting the detection of aerosol mass concentration. Directly comparing the 3 different PTR-TOF-MS used with the same integration times would provide LODs within the same order of magnitude. In order to harmonize the pre-concentration factors of the aerosol collecting techniques (ACM and TD) a 3 min average desorption time was assumed for an individual compound

thus a recalculation of the pre-concentration factor and therefore the LOD was performed for the ACM and TD and has been updated in table 2 and throughout the manuscript.

A discussion was added in section 2.3 by inserting at line 165 "The pre-concentration factor for ACM and TD was calculated from the ratio of the volume sampled during collection to the volume evaporated during desorption, assuming a 3 min desorption time for an individual compound." And at line 168 "It should be noted that for the individual PTRMS the LOD for gas-phase measurements, bypassing any pre-concentration step, agreed within a factor of two."

**(4)** The paper describes the aerosol sampling used in the three instruments in great detail, which is appropriate. However, there is very little detail about the PTR-TOF-MS systems used. What were the types of instruments used, why is the mass resolution so different between the three systems (Table 2) and how did the primary ion signals and calibration factors compare between the three systems?

Additional information is added in Table 2 and a detailed discussion is provided in section 2.3 by inserting at line 175 "An overview of the primary ion distribution is provided in Figure S 1. Normalization of the signal was performed based on the sum of 500 * $H_3O^+$ + 250 * $H_3O^+(H_2O)$ for all PTRMS. ACM and TD showed more than 98 % of the primary ions originating from $H_3O^+$ while for CHARON, when operated at 100 Td (1 Td = $10^{-17}$ V cm$^{-2}$ molecule$^{-1}$), around 65% originated from $H_3O^+$ and 35% from $H_3O^+(H_2O)$, and for CHARON at 65 Td, around 20% from $H_3O^+$ and 75% from $H_3O^+(H_2O)$." and at line 178 "All PTR-ToF-MS used in this campaign were of the model PTR-TOF 8000, manufactured from Ionicon Analytik GmbH, Innsbruck, Austria. Although originating from the same model, minor differences in the design e.g. the TOF interface existed, related mostly to ACM when compared to CHARON and TD. These differences introduced additional fragmentation and affected the resolution of the PTRMS as reflected from Table 2. Nevertheless, the sensitivity of all PTRMS when using acetone as a common calibration compound was in a similar range as observed in Figure S1. When calculating the sensitivity using the cps instead of the ncps, observed differences suggested lower primary ion signal and reaction times for ACM and TD when compared to CHARON. In the following subsections the principle of operation and operating conditions of the different inlets and PTRMS systems used in this study is reported."

**Detailed comments:**
**(5)** Line 34: "predominantly" instead of "predominately"
Done

**(6)** Line 37: "carbon-oxygen bond breakage" appears to be used here and throughout the text as synonymous with process that lower the O:C ratio. However, carbon-oxygen bonds are not necessarily broken in all fragmentation processes, so I would recommend the more general "fragmentation".
"Carbon-oxygen bond breakage" was changed to "fragmentation" throughout the manuscript.

**(7)** Line 88: "low-volatility VOCs" instead of "low VOCs"?
Done

**(8)** Table 2: Please add the temperature and pressure of the drift tube reactors used in these experiments. Also useful would be more details on the specific TOF-MS systems used and how these

translate into the primary ion signals and sensitivities (in raw and/or normalized counts per seconds) of the three systems.

The temperature and pressure of the drift tube were added in Table 2. For the rest see comment (4)

**(9)** Lines 165-168: Limits of detection vary by orders of magnitude between the three instruments. Part of the difference (between TD and CHARON) must be related to the time response of the methods? How do the detection limits compare if the same averaging times are used? The scatter in Figure 1 appears to indicate that the precision of the measurements is similar when averaged over the same time, but perhaps the data should not be interpreted like that.

See comment (3)

**(10)** Line 168: "V/cm" instead of "V cm"

Done

**(11)** Lines 168-169: A graph showing the different distributions of primary ions in the three different instruments would be helpful.

See comment (8)

**(12)** Line 183: The mass resolution is quoted as 2500 in Table 2. Why is there such a large difference with the specifications of the mass spectrometer?

See comment (4)

**(13)** Figure 1b: the error bars and line fitted through the data points are hard to see.

Done

**(14)** Line 348: "variability" instead of "uncertainties"?

Done

**(15)** Lines 351-354: how about incomplete evaporation of the sampled OA? The upper temperature in the three instruments is quite different and some, like the CHARON, appear to be low compared to other thermal desorption measurements. For example, Figure S1 shows that an appreciable amount of OA evaporates above 150 C. How about transmission losses of OA vapors from the evaporation zone to the drift tube?

This is a very good point. Discussions were added in section 3.1 inserting the incomplete evaporation or transmission as an additional source of losses by adding at line 383 "The thermal desorption process varied for the different PTR-based inlet techniques with different desorption residence times, desorption temperatures and pressure conditions (see section 2.3). Although CHARON was operated at lower temperatures compared to ACM and TD, its reduced pressure compensated for the temperature difference thus increasing the volatility range down to LVOC (Eichler et al., 2017). It could still be though that a fraction of the SOA mass in the extremely low volatility OC (ELVOC) range will not evaporate during desorption from any of the systems studied.  If this effect would be significant it would be more pronounced in the presence of high percentages of ELVOCs in the aerosol, i.e. during periods with

increased O:C ratios (indicated in Figure 2). A non-linear relationship between SMPS and the PTR based techniques would be the result, which has not been observed (Figure 1). We therefore concluded that incomplete evaporation of ELVOC constitutes a minor contribution to the mass recovery underestimation. Transmission losses of OA vapours on the pathway from evaporation to detection could occur on cold spots in between the evaporation zone and the drift tube. All components were heated to higher temperatures than the evaporation zone in order to avoid these losses. Within the drift tube of the PTR the temperature is lower than in the evaporation zone but the lower pressure will reduce but not exclude the possibility of re-condensation of organic vapours."

**(16)** Lines 417-429: by assuming that the AMS gives the correct elemental composition of the OA (a big if), is it possible to derive stochastically what the average composition of the undetected fragments in the CHARON measurements is?

As correctly mentioned from the referee AMS provides the elemental composition of the OA after applying correction factors that introduce uncertainty. Although a stochastic calculation of the average composition of the undetected fragments could be performed the outcome will be highly uncertain. Future studies focusing on single compound systems could provide more reliable insights to this question but are beyond the scope of an instrumental comparison of PTR-based techniques.

**(17)** Figure 4: these graphs might be a little easier to look at, if the ACM and TD results were shown next to each other, instead of having the CHARON results in between.

This graph focuses on the residual of ACM and TD to CHARON. The reason CHARON was chosen to be in the center was in order to emphasize these differences. For this reason we decided to keep the graph in the same order.

**Response to Referee #2**

**(1)** PTR-ToF-MS provides real-time, robust measurements of ambient VOCs. This manuscript expands the use of PTR-ToF-MS applications to include particulate bound organics and compares performances of three different aerosol sampling techniques, aerosol collection module (ACM), the chemical analysis of aerosol online (CHARON), and the thermal desorption (TD) to evaluate their ability to provide chemical details of organic aerosol when coupled with PTR-ToF-MS. It also examines their ability to provide additional information relevant to the organic aerosol such as aging, O:C ratios, and volatility patterns. The authors performed carefully designed experiments to replicate the formation and aging of biogenic SOA and did careful analyses and interpretation of the results considering different factors that can affect the experimental results as E/N. Considering the importance of organic aerosols in the atmosphere and the difficulties associated with the chemical characterization of organic aerosols, this work is valuable as it expands and improves the atmospheric measurements techniques for organic aerosol speciation. Therefore, I recommend this work for publication in AMTD after minor revisions.

We thank the referee for the useful comments. All revisions have been accounted for as described in the following.

**(2)** Although this work aimed to compare performances of different aerosol sampling technique, the operating conditions and PTR-ToF-MS setups were not the same for three aerosol samplers, which affected the measured collected efficiency. The authors discuss the effect of E/N on the ionic fragmentation in the drift tube at the end of this discussion. I would move this fact to the beginning of the discussion in section 3 so there is no suspense and modify the figure captions to include the different operating conditions.

The structure of section 3 follows the path of the particles from sampling to detection with their respective characteristics. This was the main reason E/N was introduced last although having a strong influence on the fragmentation patterns.

We made a comment in line 353 adding "The extent to which these processes affect the different techniques was investigated in detail and presented in the following by tracking the path of the particles from collection to detection."

**(3)** The authors compare the organic mass concentration corresponding to different aerosol samplers and AMS to that of SMPS. These instruments measure particles with different size ranges. There is no discussion of aerosol size distribution. It is worth to include a short discussion on measured particle size distribution and samplers' size range. Also it is not certain why the authors compared the organic mass measurements by the three samplers to that of SMPS, which is derived using a density correction. Would not this be more reliable to compare those to AMS derived mass concentration? Also this comparison implies that the aerosol volume measured by SMPS is 100% OC. Is it correct assumption?

This is a good point. We added in the supplementary Figure S3 presenting the volume size distribution measured from the SMPS. A discussion is added in section 3.1 by inserting in line 340 "Each aerosol technique was collecting/detecting particles in different size ranges (Table 2). The volume distribution derived from SMPS measurements (Figure S3) covered a particle diameter range of 100 to 400 nm which is within the size detection limits of all applied aerosol techniques."

As discussed in the manuscript AMS suffers from CE losses. The usual approach in order to correct AMS data for CE is by applying a correction factor obtained based on the SMPS data (Docherty et al., 2013). Since SMPS is the most reliable technique in regard to particle detection compared to all other techniques used in this campaign this was the main reason we used it as the reference instrument. Concerning the density assumption a discussion is added in section 3.1 by inserting in line 340 "SMPS organic mass concentration was calculated assuming a density of 1.4 g cm$^{-3}$, a valid assumption for SOA (Cross et al., 2007), that represented more than 98 % of the mass, as observed from AMS."

**(4)** As the organic aerosol age, more volatile smaller chain oxygenates can gas off the aerosol surface, thus affecting the O:C ratio and volatility. The authors discussed effect of PTR measurement conditions on the fragmentation, but not much about the fragmentation/ gassing off due to oxidation of OA. Can the authors comment or include a discussion?

A characterization of the aerosol phase such as aging is beyond the scope of this publication which deals with the inter-comparison of four different aerosol measurement techniques. A separate publication in preparation will focus on the gas-to-particle partitioning and address the issue of volatility and its dependence on the O:C ratio.

**(5)** Although the manuscript is structured well, the language and writing could be improved. It is recommended the authors do a thorough proofreading and improve the fluency. Few examples include:

**Page 5 line 147:** replace "where" with "was"

Done

**Page 12 line 435:** . . .ratios were lower that . . . instead of ". . .. . .ratios was lower that. . .. .. .. . .. . .."

Done

**Page 15 line 527-530:** consider re-phrasing.

Done

**Page 17 line 610:** "aging" instead of "ageing"

Done

**References**

Cross, E. S., J. G. Slowik, P. Davidovits, J. D. Allan, D. R. Worsnop, J. T. Jayne, D. K. Lewis, M. Canagaratna, and T. B. Onasch: Laboratory and Ambient Particle Density Determinations using Light Scattering in Conjunction with Aerosol Mass Spectrometry, *Aerosol Science and Technology*, *41*(4), 343-359, doi:10.1080/02786820701199736, 2007.

Docherty, K. S., M. Jaoui, E. Corse, J. L. Jimenez, J. H. Offenberg, M. Lewandowski, and T. E. Kleindienst: Collection Efficiency of the Aerosol Mass Spectrometer for Chamber-Generated Secondary Organic Aerosols, *Aerosol Science and Technology*, *47*(3), 294-309, doi:10.1080/02786826.2012.752572, 2013.

Eichler, P., M. Müller, C. Rohmann, B. Stengel, J. Orasche, R. Zimmermann, and A. Wisthaler: Lubricating Oil as a Major Constituent of Ship Exhaust Particles, *Environmental Science & Technology Letters*, *4*(2), 54-58, doi:10.1021/acs.estlett.6b00488, 2017.

Holzinger, R., A. Kasper-Giebl, M. Staudinger, G. Schauer, and T. Röckmann: Analysis of the chemical composition of organic aerosol at the Mt. Sonnblick observatory using a novel high mass resolution thermal-desorption proton-transfer-reaction mass-spectrometer (hr-TD-PTR-MS), *Atmospheric Chemistry and Physics*, *10*(20), 10111-10128, doi:10.5194/acp-10-10111-2010, 2010.

**Table 2: Instruments operating conditions.**

| INSTRUMENT CHARACTERISTICS | ACM (in situ) | CHARON (online) | TD (in situ) |
|---|---|---|---|
| **Time resolution (min)** | 240 | 1 | 120 |
| **Gas/particle separation** | High vacuum | Denuder | Denuder and/or blank correction (filtered air) |
| **Pre-concentration factor** | 21[a] | 44 | 6000[b] |
| **LOD[c] (ng/m$^3$)** | 35[d] | 1.4[e] | 0.02[b] |
| **Temperature range (°C)** | 25 – 250 | 140 | 25 – 350 |
| **Heating rate (°C / min)** | 100 | 0 | 15 |
| **Temperature steps (°C)** | 100, 150, 250 (3 min) | none | None |
| **Desorption pressure (atm)** | 1 | < 1 | 1 |
| **Particle range (nm)** | 70 – 1000 | 70 – 1000 | 70 - 2000 |
| **PTR-ToF-MS model** | 8000 | 8000 | 8000 |
| **Drift tube Temperature (°C) / Pressure (mbar) / Voltage (V)** | 90 / 2.3 / 550 | 120 / 2.4 / 400 and 240 | 120 / 2.25 / 600 |
| **PTR-ToF-MS E/N (Td)** | 120 | 65 / 100 | 160 |
| **PTR-ToF-MS mass resolution (m/Δm)** | 2500 | 4500-5000 | 4000 |

[a] based on 240 min sampling at 80 mL/min and 3 min desorption at 300 mL/min

[b] based on 30 min sampling at 6 L/min and 3 min desorption at 10 mL/min a typical value for most ions based on the method in (Holzinger et al., 2010)

[c] Limit of detection

[d] For signal on *m/z* 139 and 10 sec integration time

[e] For signals around *m/z* 200 and 1 min integration time

[Figure]

**Figure S 1: (a)** The normalized primary ion distribution as observed for the different PTR-based techniques operated at different E/N conditions and **(b)** the sensitivity of acetone both in counts per second (cps) per ppbV and normalized cps (ncps) per ppb for each instrument.

[Figure]

**Figure S 3: The volume size distribution measured from an SMPS during the (i) β-pinene, (ii) limonene, (iii) β-pinene/limonene mixture and (iv) tree emissions oxidation experiments.**

[Figure]

**Figure S 4:** The time series of (a) the particulate organic mass concentration (left axis) and nitrate mass concentration (right axis) in µg m$^{-3}$ and (b) the gas-phase NO (left axis) and NO$_2$ (right axis) mixing ratios in ppbV throughout the campaign. Information on the type of precursor experiment performed is provided above the graph together with indications for periods of the chamber illumination (yellow background color) and NO$_3$ oxidation (blue background color). The maximum organic nitrate fraction can be estimated from the measurement of the total nitrate derived by AMS. Adding an organic backbone to the nitrate with a maximum molecular weight of 180 g mol$^{-1}$ results in a total organic nitrate concentration of M(NO$_3^-$+ Org)/M(NO$_3^-$)* C(NO$_3^-$)$_{max}$ = (62+180)/(62)* 0.8 = 3.1 µg m$^{-3}$ which corresponds to a maximum of 10% for the limonene NO$_3$ oxidation.